# PCAF-mediated acetylation of ISX recruits BRD4 to promote epithelial-mesenchymal transition

Li-Ting Wang[1], Kwei-Yan Liu[2,3], Wen-Yih Jeng[4,5], Cheng-Ming Chiang[6], Chee-Yin Chai[7], Shyh-Shin Chiou[3,8,9,10], Ming-Shyang Huang[11], Kazunari K Yokoyama[1,12,13] (iD), Shen-Nien Wang[1,14,15,16], Shau-Ku Huang[2,17,18] & Shih-Hsien Hsu[1,3,16,19,*] (iD)

## Abstract

Epigenetic regulation is important for cancer progression; however, the underlying mechanisms, particularly those involving protein acetylation, remain to be fully understood. Here, we show that p300/CBP-associated factor (PCAF)-dependent acetylation of the transcription factor intestine-specific homeobox (ISX) regulates epithelial–mesenchymal transition (EMT) and promotes cancer metastasis. Mechanistically, PCAF acetylation of ISX at lysine 69 promotes the interaction with acetylated bromodomain-containing protein 4 (BRD4) at lysine 332 in tumor cells, and the translocation of the resulting complex into the nucleus. There, it binds to promoters of EMT genes, where acetylation of histone 3 at lysines 9, 14, and 18 initiates chromatin remodeling and subsequent transcriptional activation. Ectopic ISX expression enhances EMT marker expression, including TWIST1, Snail1, and VEGF, induces cancer metastasis, but suppresses E-cadherin expression. In lung cancer, ectopic expression of PCAF–ISX–BRD4 axis components correlates with clinical metastatic features and poor prognosis. These results suggest that the PCAF–ISX–BRD4 axis mediates EMT signaling and regulates tumor initiation and metastasis.

**Keywords** BRD4; EMT; ISX; TWIST1

**Subject Categories** Cancer; Chromatin, Transcription & Genomics; Post-translational Modifications & Proteolysis

See also: **MP Stemmler** (February 2020)

## Introduction

Epigenetic regulation has been broadly defined as alteration of gene expression through chromatin structure modification without changing the underlying nucleotide sequences [1]. Histone acetylation, a type of epigenetic modification, plays an important role in gene regulation during embryonic development and human disease progression [2]. Many histone acetyltransferases (HATs), including p300/CBP-associated factor (PCAF), p300/CREB binding protein (CBP), TIP60, and hMOF, are involved in histone acetylation, which has been linked to transcriptionally active chromatin and subsequent gene transcription [2,3]. Studies have also suggested the involvement of HAT dysregulation in cancer progression, especially in cancer metastasis and recurrence [3,4]. PCAF, a member of the GCN5-related N-acetyltransferase family of protein acetyltransferases, has been shown to be involved in the modulation of differentiation, angiogenesis, cell cycle progression, gluconeogenesis, and carcinogenesis; however, the pathological functions of PCAF in cancer progression remain controversial [5,6]. Through its ability to

1   Graduate Institute of Medicine, College of Medicine, Kaohsiung Medical University, Kaohsiung, Taiwan
2   Department of Respirology & Allergy, Third Affiliated Hospital of Shenzhen University, Shenzhen, China
3   Center of Applied Genomics, Kaohsiung Medical University, Kaohsiung, Taiwan
4   University Center for Bioscience and Biotechnology, National Cheng Kung University, Tainan, Taiwan
5   Department of Biochemistry and Molecular Biology, National Cheng Kung University, Tainan, Taiwan
6   Department of Biochemistry, and Department of Pharmacology, Simmons Comprehensive Cancer Center, University of Texas Southwestern Medical Center, Dallas, TX, USA
7   Department of Pathology, Faculty of Medicine, Kaohsiung Medical University, Kaohsiung, Taiwan
8   Department of Pediatrics, Faculty of Medicine, College of Medicine, Kaohsiung Medical University, Kaohsiung, Taiwan
9   Division of Hematology-Oncology, Department of Pediatrics, Kaohsiung Medical University Hospital, Kaohsiung Medical University, Kaohsiung, Taiwan
10  Research Center for Environmental Medicine, Kaohsiung Medical University, Kaohsiung, Taiwan
11  Department of Internal Medicine, E-Da Cancer Hospital, School of Medicine, I-Shou University, Kaohsiung, Taiwan
12  Center of Stem Cell Research, Kaohsing Medical University, Kaohsing, Taiwan
13  Graduate Institute, The University of Tokyo, Tokyo, Japan
14  Division of General and Digestive Surgery, Department of Surgery, Kaohsiung Medical University Hospital, Kaohsiung, Taiwan
15  Department of Surgery, College of Medicine, Kaohsiung Medical University Hospital, Kaohsiung, Taiwan
16  Center for Cancer Research, Kaohsiung Medical University, Kaohsiung, Taiwan
17  National Institute of Environmental Health Sciences, National Health Research Institutes, Zhunan, Taiwan
18  Johns Hopkins University School of Medicine, Baltimore, MD, USA
19  Department of Medical Research, Kaohsiung Medical University Hospital, Kaohsiung Medical University, Kaohsiung, Taiwan
    *Corresponding author. Tel: +886 7 3121101; E-mail: jackhsu@kmu.edu.tw

interact with p300/CBP, PCAF forms a multimeric acetylase complex that remodels chromatin and facilitates downstream gene expression [7]. Moreover, PCAF acetylates not only histones to promote gene transcription but also certain non-histone transcription factors (TFs), such as p53 and STAT3, to directly promote their transcriptional activity [6–8]. Evidence suggests that PCAF functions as a key regulator of these non-histone proteins, which coordinate many carcinogenic and tumor suppression processes, such as cell cycle progression, DNA damage response, and apoptosis [9]. Although PCAF exerts important effects on the functions of nucleosomes and TFs, the underlying mechanisms of these effects in the cytosol and nuclei remain largely unknown.

Bromodomain-containing protein 4 (BRD4), a member of the bromodomain and extraterminal (BET) protein family, plays an important regulatory role in early embryonic development and human disease progression [10]. A recent report demonstrated that BRD4 controlled tumor metastasis via stability and expression of Snail in breast cancer [11]. Blocking BRD4 interactions by small-molecule inhibitors has been shown to effectively inhibit cell proliferation in cancers [12,13], and many potential candidates have been devoted to human clinical trials. As an organizer of super-enhancers (SEs) of hyper-acetylated promoter nucleosomes through a bromodomain-mediated recruitment mechanism, BRD4 has been shown to interact with TFs that facilitate downstream gene expression [14,15], including oncogenes [12]. Moreover, histone acetylation at H3 lysine residues 9 and 27 was found to be important in BRD4-mediated SE-associated nucleosome organization [12,16]. Accordingly, the loss of BRD4 abolished enhancer-mediated gene expression. Although genome-wide studies have indicated that BRD4 is widely distributed along the genome, selective gene expression patterns and how recruitment of HAT-containing co-activators to specific promoters modulates TF-associated nucleosome organization remain largely undefined.

The present study shows that PCAF acetylation of intestine-specific homeobox (ISX) recruits BRD4 to promoter nucleosome movement at EMT initiator, such as TWIST1 and Snail1, thereby facilitating chromatin remodeling and transcriptional initiation via histone H3 acetylation. Enhanced expression of EMT initiators leads to tumor microenvironment remodeling, thereby promoting metastasis *in vitro* and *in vivo*. Ectopic expression of PCAF–ISX–BRD4 axis components correlates with clinical metastatic features and poor prognosis, suggesting that the PCAF–ISX–BRD4 axis is an important regulator of tumor metastasis and cell plasticity in a tumorigenic microenvironment.

# Results

## ISX transactivates EMT regulators and promotes EMT characteristics

To study the potential effects of ISX on poor prognosis among patients with cancer [17,18], EMT regulator expression as well as EMT characteristics were initially evaluated in cancer cells expressing ISX. Lung cancer cells (A549 and H1299) showed increased mRNA and protein levels of mesenchymal cell markers, such as TWIST1, Snail, Slug, ZEB1, Bmi1, fibronectin, N-cadherin, vimentin, and VEGF, after ISX induction using doxycycline (Dox.; 1 μg/ml) but decreased expression of epithelial cell marker E-cadherin (Fig 1A

and B). Promoter analysis showed that ISX-GFP transactivated luciferase activity driven by Snail1 and TWIST1 promoters (−71 to −87, and −75 to −89 bp, respectively, relative to the transcription start site) in A549 cells (Fig 1C and D). Moreover, ISX-GFP showed high binding activity to the above promoter elements of both Snail1 and TWIST1 with ISX *cis*-binding element (IBE; the sequence "CGCCGC" is a potential ISX-binding cis-element; [17,19]) in lung cancer cells (Figs 1E and EV1A). Conversely, promoters without the IBE showed no or lesser promoter activity and binding than that with GFP expression control (Figs 1C–E and EV1A). The ISX-induced transactivation was completely abolished upon deletion of the IBE in lung cancer cells (Figs 1F and EV1B). The endogenous binding activity of ISX on promoters of EMT markers was verified by Chromatin Immunoprecipitation (ChIP), and the results showed endogenous ISX bound directly to the promoters of genes involved in EMT, and IL6 promoted the promoter-binding activity of ISX (Fig 1G). The migration and invasion characteristics of ISX in lung cancer cells were further accessed using wound-healing and Transwell invasion assays. As predicted from the previous results, cells with forced ISX expression had higher migration and invasion activities compared with mock-transfected A549 lung cancer cells. However, cancer cells with ISX knockdown did not exhibit enhanced cell migration and invasion induced by ISX (Fig 1H and I). These results suggested that ISX expression transcriptionally upregulated EMT regulators and promoted EMT characteristics in lung cancer cells.

## PCAF modulated EMT characteristics induced by the ISX-BRD4 complex *in vitro* and *in vivo*

To elucidate the transactivation mechanisms of ISX on EMT regulators, co-immunoprecipitation coupled with two-dimensional gel electrophoresis (2-DE) and liquid chromatography–mass spectrometry was conducted to identify ISX-interacting proteins in lysates of the aggressive lung cancer cell line A549. This approach yielded 11 candidate proteins from three independent 2-DE experiments. One of the oligopeptides, $NH_2$-KADTTTPTTIDPIHEPPSLPPEPK-COOH, from differential expression protein was sequenced via liquid chromatography–mass spectrometry and was identified as BRD4 (gi 71052031; amino acids 291–314 on BRD4) (Fig 2A). To confirm the interaction between ISX and BRD4 *in vivo*, immunoprecipitates from A549 and H1299 cells obtained using anti-ISX polyclonal antibody were pulled down and examined by immunoblotting. As predicted, BRD4 was detected in ISX immunoprecipitates, whereas ISX was detected in anti-BRD4 immunoprecipitates (Figs 2B and EV1C). Interestingly, PCAF, an acetylation regulator, was detected in both ISX and BRD4 immunoprecipitates. Moreover, analysis of six lung cancer cell lines (A549, H358, H441, H1299, H1435, and H1437) revealed that the mRNA and protein expression patterns of ISX, PCAF, and BRD4 were co-expressed in lung cancer cells relative to those in human diploid lung fibroblasts (WI38; Fig EV1D and E). The interaction between ISX, PCAF, and BRD4 was further evaluated in tumors and adjacent healthy lung tissues from patients with lung cancer. A significant amount of BRD4 and PCAF was detected in anti-ISX immunoprecipitates of lung tumor tissues from patients with non-small-cell lung carcinoma (NSCLC), whereas low levels of BRD4, but not PCAF, RNA pol II, and CBP/CREB, were detected in anti-ISX immunoprecipitates of adjacent healthy lung tissues from patients with lung cancer (Fig 2C). Confocal fluorescence imaging

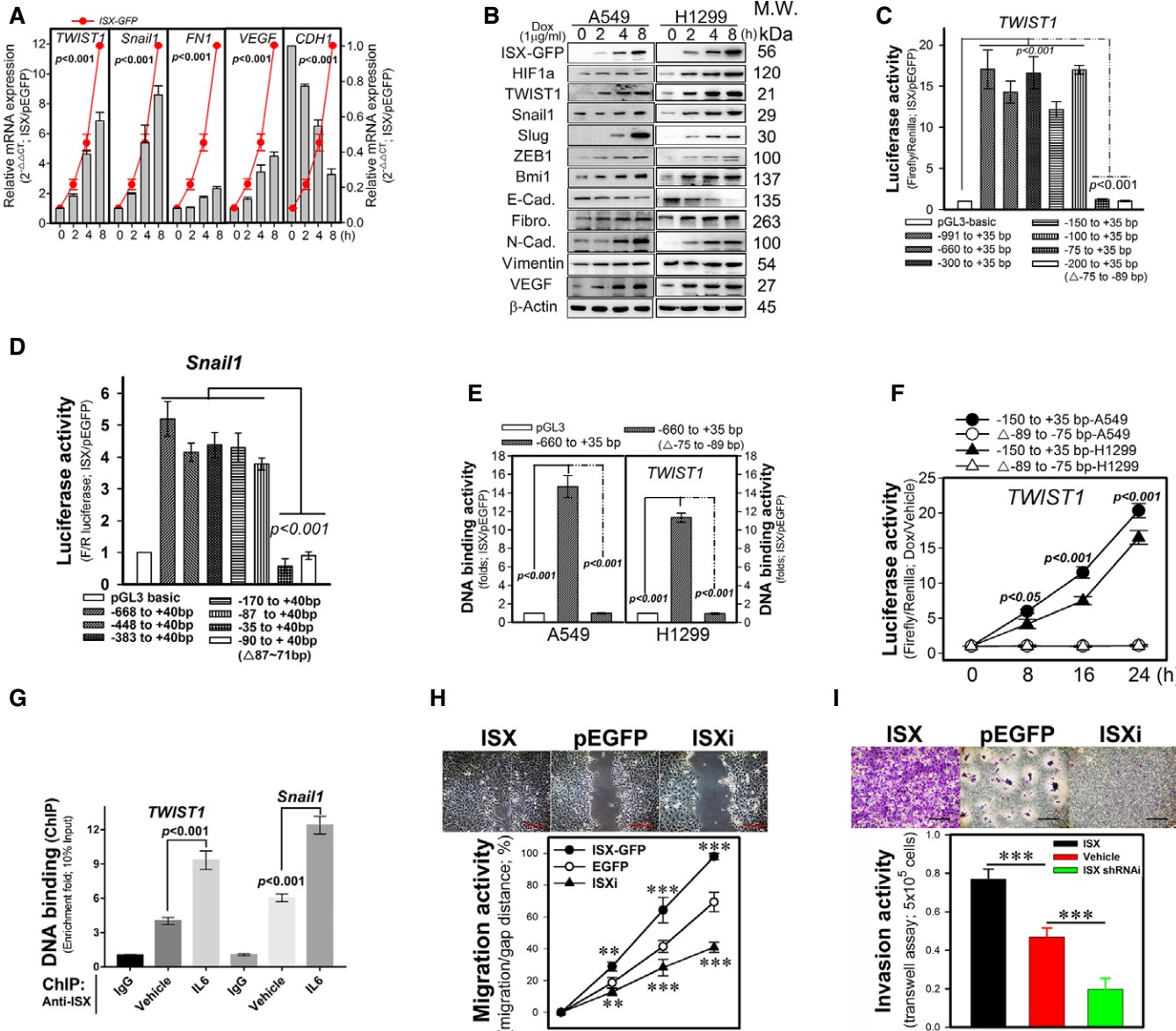

**Figure 1. ISX expression induces TWIST1 and Snail1 expression, and promotes EMT.**

A   The expression levels of *ISX*, *TWIST1*, *Snail1*, fibronectin (*FN1*), *VEGF*, and E-cadherin (*CDH1*) mRNA were examined in a ISX-GFP inducible Tet-ON transformants at 8 h after addition of DOX. Data are presented as mean ± SD in graph (*P* < 0.001 compared with time point 0 h; Student's *t*-test) of three independent experiments, each performed in triplicate.

B   Western blotting analysis of the protein levels of ISX, HIF1α, TWIST1, Snail1, Slug, ZEB1, Bmi1, E-cadherin (E-Cad.), fibronectin, N-cadherin (N-Cad.), vimentin, and VEGF in A549 and H1299 cells with DOX-inducible ISX expression system 8 h after DOX induction.

C, D   ISX transcriptionally activates luciferase activity driven by TWIST1 (C) and Snail1 (D) promoter regions in A549 cells. Data are presented as mean ± SD in bar graph (*P* < 0.001, Student's *t*-test) of three independent experiments, each performed in triplicate.

E   ChIP analysis of ISX binding to the TWIST1 promoter region in A549 and H1299 cells (10% input of each group was pull down and applied to qPCR). Data are presented as mean ± SD in bar graph (*P* < 0.001, Student's *t*-test) of three independent experiments, each performed in triplicate.

F   ISX transactivation activity analyzed by luciferase activity driven by the TWIST1 promoter. Data are presented as mean ± SD in graph (*P* < 0.001, Student's *t*-test) of three independent experiments, each performed in triplicate.

G   ChIP analysis of ISX binding to the endogenous promoters of TWIST1 (−180/+35) and Snail1(−160/+40) in A549 and cells (10% input of each group was pull down and applied to qPCR). Data are presented as mean ± SD in bar graph (*P* < 0.001, Student's *t*-test) of three independent experiments, each performed in triplicate.

H   Effect of forced expression and knockdown of ISX on cell migration (wound healing) measured in A549 cells. ISXi, ISX-specific shRNAi (described in Materials and Methods). Data are presented as mean ± SD in graph (**P* < 0.01, ***P* < 0.001, Student's *t*-test) of three independent experiments, each performed in triplicate. Scale bar, 100 μm.

I   Effect of forced expression and knockdown of ISX on cell invasion (Transwell) activity measured in A549 cells. Data are presented as mean ± SD in bar graph (****P* < 0.001, Student's *t*-test) of three independent experiments, each performed in triplicate. Scale bar, 100 μm.

Source data are available online for this figure.

and proximity ligation assay (PLA) were then used to examine the interaction between ISX, PCAF, and BRD4 in A549 cells. Results showed that both BRD4 (red) and PCAF (pink) proteins were co-localized with ISX (green) in the cytosol and nuclei (blank arrow) of A549 cells, and PLA analysis showed positive PLA signals (red) for both BRD4 and PCAF with ISX (Figs 2D and EV1F).

To determine the potential effects of PCAF on EMT characteristics induced by ISX, A549 cells with forced ISX expression were treated with four acetyltransferase inhibitors to evaluate the potential regulatory effects of PCAF on ISX–BRD4 complex formation and EMT characteristics induced by ISX. Garcinol (p300 and PCAF inhibitor), C646 (p300/CBP inhibitor), and MB–3 (GCN 5 inhibitor), but

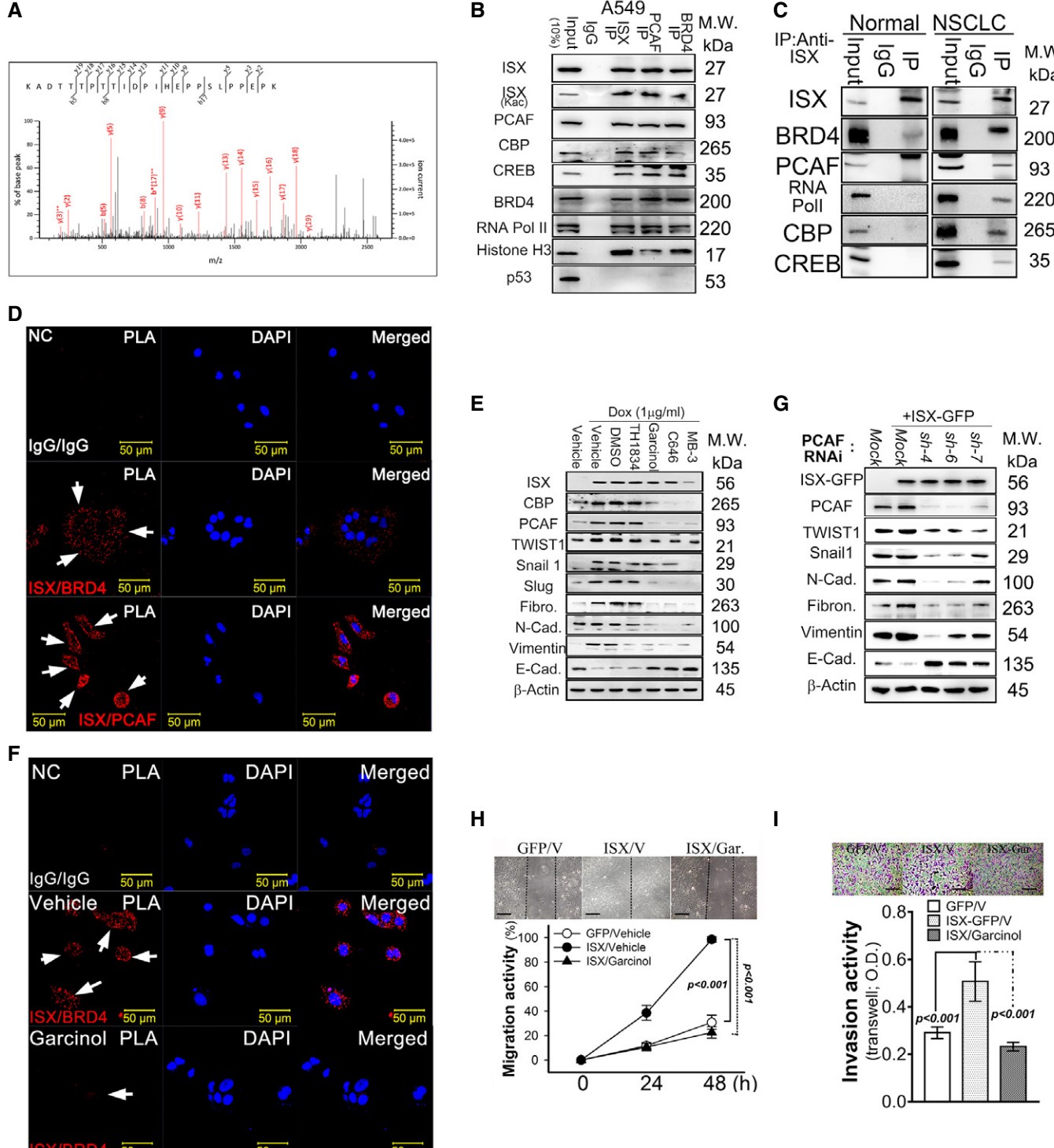

**Figure 2.**

**Figure 2.　ISX interacts with both BRD4 and PCAF *in vitro* and *in vivo*.**

A　　BRD4 peptide (291–314 a.a) obtained from anti-ISX immunoprecipitates of A549 cell lysates identified through liquid chromatography–tandem mass spectrometry (LC-tandem-MS).

B　　ISX association proteins (PCAF, BRD4, CBP, CREB, RNA Pol II, and Histone H3) determined by Western blot in immunoprecipitation of A549 cells.

C　　ISX association proteins (PCAF, BRD4, CBP, CREB, and RNA Pol II) determined by Western blot in anti-ISX immunoprecipitates of tumor tissues from patients with lung cancer.

D　　Proximity ligation assay (PLA) of ISX and BRD4(PCAF) interaction in A549 cells. Red foci indicate close proximity of the two proteins. NC: negative control. Blank arrows indicate interactions of ISX with BRD4 (PCAF) in the cells.

E　　Western blotting analysis of the protein levels of ISX, PCAF, and EMT markers in DOX-inducible ISX expression system A549 cells treated with TH1834, Garcinol, C646, and MB-3 for 8 h after DOX induction.

F　　Proximity ligation assay of ISX and BRD4 interactions in A549 cells treated with Garcinol. Red foci indicate close proximity of the two proteins. Blank arrows indicate interactions of ISX with BRD4 in the cells. NC: negative control.

G　　Western blotting analysis of the protein levels of ISX, PCAF, and EMT markers in A549 cells with PCAF knockdown and DOX-inducible ISX expression system at 8 h time point after DOX induction.

H, I　　Cell migration (wound healing, h) and invasion (Transwell, I) assay of A549 lung cancer cells treated with Garcinol. V, vehicle. Data are presented as mean ± SD in graph (***$P < 0.001$, Student's *t*-test) of three independent experiments, each performed in triplicate. Scale bar, 100 μm.

Data information: Each experiment was repeated at least three times.
Source data are available online for this figure.

---

not TH1834 (TIP60 inhibitor), were found to abrogate the enhanced expression of EMT regulators (TWIST1, Snail1, and Slug) and markers (fibronectin, N-cadherin, and vimentin) induced by forced ISX expression (Fig 2E). Moreover, the expression of ISX (green) and most BRD4 (red) showed a cytosol localization pattern in A549 cells, while garcinol treatment of the cells abolished PLA signals, as determined through confocal immunofluorescence imaging (Figs 2F and EV1G and H). Also, the ISX-induced expression of EMT markers was abrogated in cells with PCAF knockdown (Figs 2G and EV1I). Further, A549 cells with forced ISX expression showed significantly decreased cell migration and Transwell invasion after garcinol treatment (Fig 2H and I). The above results thus suggest that acetylation by the p300/CBP/PCAF complex modulates the expression of EMT regulators and EMT characteristics induced by forced ISX expression.

**PCAF acetylation of ISX at lysine residue 69 is essential for ISX–BRD4 complex formation and induces EMT and cancer cell metastasis**

To further determine the effect of PCAF on ISX-induced EMT, potential PCAF acetylation motifs [6] on ISX were identified, and three point mutations of ISX at positions 38 (AC1), 69 (AC3), and 72 (AC2) were made to examine their impact on the ISX–BRD4 complex formation and EMT regulation (Fig 3A and B). Recombinant PCAF

protein significantly acetylated wild-type ISX, AC1, and AC2 mutant proteins. However, PCAF showed no acetylation activity on recombinant ISX mutant protein at position 69 (AC3) *in vitro* (Fig 3C). Acetylated wild-type recombinant ISX was then digested with trypsin and sequenced using liquid chromatography–mass spectrometry. The peptide of ISX (NH$_2$-SDMDRPEGPGEEGPGEAAASGSGLEKPPK-COOH, amino acids 44–72) was identified with acetylation lysine at position 69 (y(4): 469.31–511.31 *m/z*; Fig EV2A and B). A549 and H1299 cells were then transfected with ISX mutants and the expression level and localization of ISX mutants, PCAF, and BRD4 were determined by immunoblotting. PCAF, BRD4, and ISX were detected both in the cytosol and in the nuclei in cells transfected with wild-type ISX, AC1, and AC2 mutants. PCAF and BRD4, as well as the ISX AC3 mutant, were mostly detected in the cytosol fraction, whereas none or low levels were detected in the nuclei of cells transfected with the AC3 ISX mutant (Fig 3D). Compared with A549 cells transfected with AC1 or AC2 ISX mutants, no or low levels of PCAF and BRD4 proteins were detected in anti-GFP immunoprecipitates of cells transfected with the ISX AC3 mutant *in vivo* (Fig 3E). Cells transfected with AC3 showed greater suppression in the expression of EMT regulators and markers compared with cells transfected with wild-type ISX and the other AC mutants (Fig EV2C). Acetylation of histones H2, H3, and H4 was assessed in A549 cells with wild-type ISX and AC mutants. Forced expression of wild-type ISX, as well as AC1 and AC2, promoted histone H3 acetylation at positions 9, 14,

---

**Figure 3.　Acetylation of ISX at lysine 69 is critical for ISX–BRD4 association.**

A, B　　Schematic representation of the potential acetylation domain organization of ISX and its lysine mutants (AC1–AC3).

C　　Recombinant PCAF acetylates His6-ISX at lysine residue 69 by *in vitro* acetylation assay. Acetylated ISX was detected by anti-acetyl Lysine antibody.

D, E　　The protein levels of GFP-tagged WT or mutant ISX, PCAF, and BRD4 were determined in cytosol, nuclei, and anti-GFP immunoprecipitates of A549 cells by Western blotting. Acetylated ISX was detected by anti-acetyl Lysine antibody.

F　　The protein levels of total and acetylated histone H3 were determined in anti-histone H3 immunoprecipitates of A549 cells by Western blotting.

G, H　　The cell migration (wound healing, G) and invasion (Transwell, H) activity were determined in A549 cells with GFP-tagged wild or ISX mutants. Data are presented as mean ± SD in graph (***$P < 0.001$, Student's *t*-test) of three independent experiments, each performed in triplicate. Scale bar, 100 μm.

I　　Schematic representation of tumor xenograft metastasis activity of constitutively expressing RFP A549 cells transfected with wild-type or ISX AC3 mutant cDNA. Modified A549 cells were directly injected into lung to form xenograft tumor mass.

J　　Tumor xenograft metastasis activity of constitutively expressing RFP A549 cells transfected with wild-type or ISX AC3 mutant cDNA was imaged by IVIS imaging system at fifth weeks.

K　　Kaplan–Meier survival curve analysis of nude mice xenograft injected with A549 cells carrying GFP, ISX-GFP, and ISX Ac-GFP (AC3) ($n = 10$). $P = 0.0003$. *P*-values were calculated by log-rank (Mantel–Cox) test comparing the two Kaplan–Meier curves.

Data information: Each experiment was repeated at least three times.
Source data are available online for this figure.

---

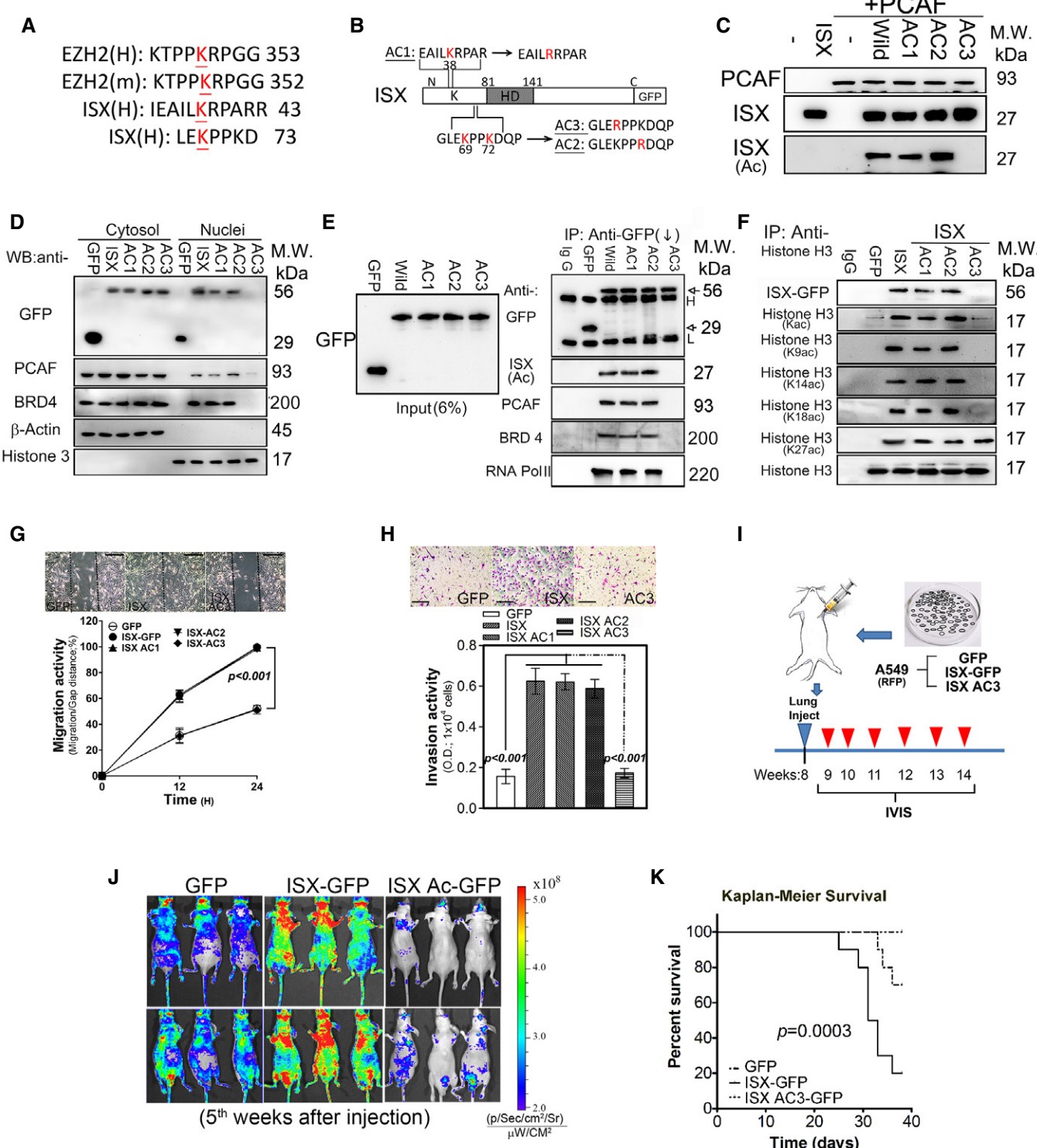

**Figure 3.**

18, and 27 (Fig 3F), whereas forced AC3 ISX mutant expression showed no histone H3 acetylation at positions 9, 14, and 18. No acetylation was detected on histones H2 and H4 with forced ISX expression (data not shown).

A549 cells transfected with AC1 or AC2 ISX mutant, as well as wild-type ISX, significantly promoted EMT characteristics [Fig 3G (migration) and h (invasion)]. However, cells transfected with AC3 ISX mutant showed no enhancement in both migration and Transwell invasion compared with cells with forced ISX or AC1 (AC2) expression. To evaluate cell migration, invasion, and metastasis *in vivo*, constitutively RFP-expressing A549 cells transfected with wild-type ISX or AC3 ISX mutant were directly injected into the

lungs of nude mice, after which an *in vivo* imaging system (IVIS) was used to monitor tumor cell progression every week (Fig 3I). Mice injected with A549 cells having forced wild-type ISX expression developed a detectable tumor at the second week in the lung and subsequent proliferation and metastasis were noted on the third week after injection. Most of mice injected with A549 cells with wild-type ISX were not survived with global tumor cell metastasis from the fourth weeks (Fig 3J and K). Conversely, A549 cells

transfected with the AC3 ISX mutant showed no or few detectable tumors at the fourth week, whereas no or minor metastases were detected at the fifth week in nude mice (Fig 3J). Nude mice injected with A549 cells expressing ISX, but not those injected with cells expressing vector or AC3 ISX, showed limited survival and died 3–6 weeks postinjection (Fig 3K). The above result showed that acetylation of ISX at lysine residue 69 is essential for ISX-BRD4 complex formation, ISX-induced EMT, and tumor metastasis in lung cancer.

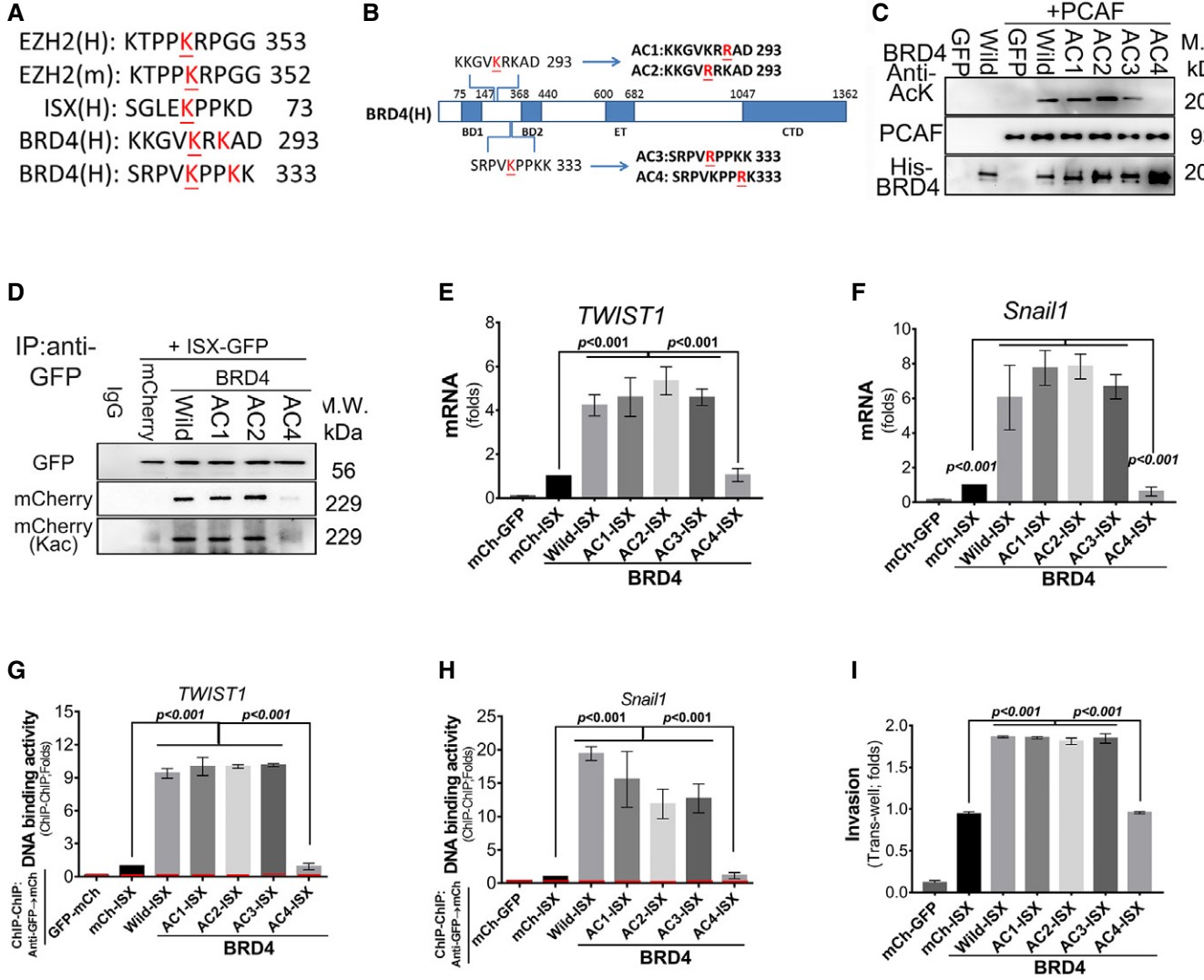

**Figure 4. Acetylation of BRD4 at lysine 332 is critical for ISX–BRD4 association.**

A, B    Schematic representation of the potential acetylation domain organization of BRD4 and its lysine mutants (AC1–AC4).

C    Recombinant PCAF acetylates His6-BRD4 at lysine residue 332. Acetylated BRD4 was detected by anti-acetyl lysine antibody.

D    The mCherry-tagged WT and BRD4 mutants were detected in anti-GFP immunoprecipitates by Western blotting in A549 cells.

E, F    The mRNA levels of TWIST1(E) and Snail1(F) were verified in A549 cells co-expressing mCherry-tagged WT or mutant BRD4 and GFP-tagged ISX by RT–PCR. Data are presented as mean ± SD in bar graph ($P < 0.001$, Student's $t$-test) of three independent experiments, each performed in triplicate.

G, H    The DNA-binding activity of TWIST1 and Snail1 was evaluated in anti-GFP-mCherry ChIP–ChIP immunoprecipitates by RT–PCR in A549 cells. Red, Hprt1 promoter (−190/+40 bp, negative control) (10% input of each group was pulled down and used for qPCR analysis). Data are presented as mean ± SD in bar graph ($P < 0.001$, Student's $t$-test) of three independent experiments, each performed in triplicate.

I    The cell invasion (Transwell) activity was determined in A549 cells co-transfected with cDNA coding for GFP-tagged ISX and mCherry-tagged BRD4 mutants. Data are presented as mean ± SD in bar graph ($P < 0.001$, Student's $t$-test) of three independent experiments, each performed in triplicate.

Data information: Each experiment was repeated at least three times.
Source data are available online for this figure.

## PCAF-induced acetylation on lysine residue 332 of BRD4 is essential for EMT activity induced by the ISX–BRD4 complex

Similarly, His$_6$-tagged wild-type and mutated BRD4 proteins were incubated with recombinant PCAF to evaluate the potential acetylation sites *in vitro* and determine whether BRD4 is a target protein of PCAF. Four potential lysine acetylation sites on BRD4 [289 (AC2), 291(AC1), 329 (AC3), and 332 (AC4)] were developed and expressed to examine the impact of the ISX–BRD4 complex on EMT in lung cancer cells (Fig 4A and B). PCAF protein showed significant acetylation with wild-type BRD4 and AC1–AC3 BRD4 mutants but not with the AC4 BRD4 mutant

(Fig 4C). Acetylated wild-type recombinant BRD4 was then digested with trypsin and sequenced by liquid chromatography–mass spectrometry. The peptide of BRD4 (NH$_2$-ESSRPVKPPKK-COOH, amino acids 323–333) was identified with acetylation lysine at position 332 (y(2): 275.21–317.21 *m/z*; Fig EV3A and B). Wild-type and mutant BRD4 were then expressed in A549 cells with ISX-GFP expression, and ectopic BRD4 proteins were detected in anti-GFP immunoprecipitates. Compared with cells transfected with wild-type or AC1–AC3 mutants, no or fewer BRD4 proteins were detected in the anti-GFP immunoprecipitates of cells transfected with the AC4 BRD4 mutant *in vivo* (Figs 4D and EV3C). Similarly, the expression of AC4 BRD4 mutant in

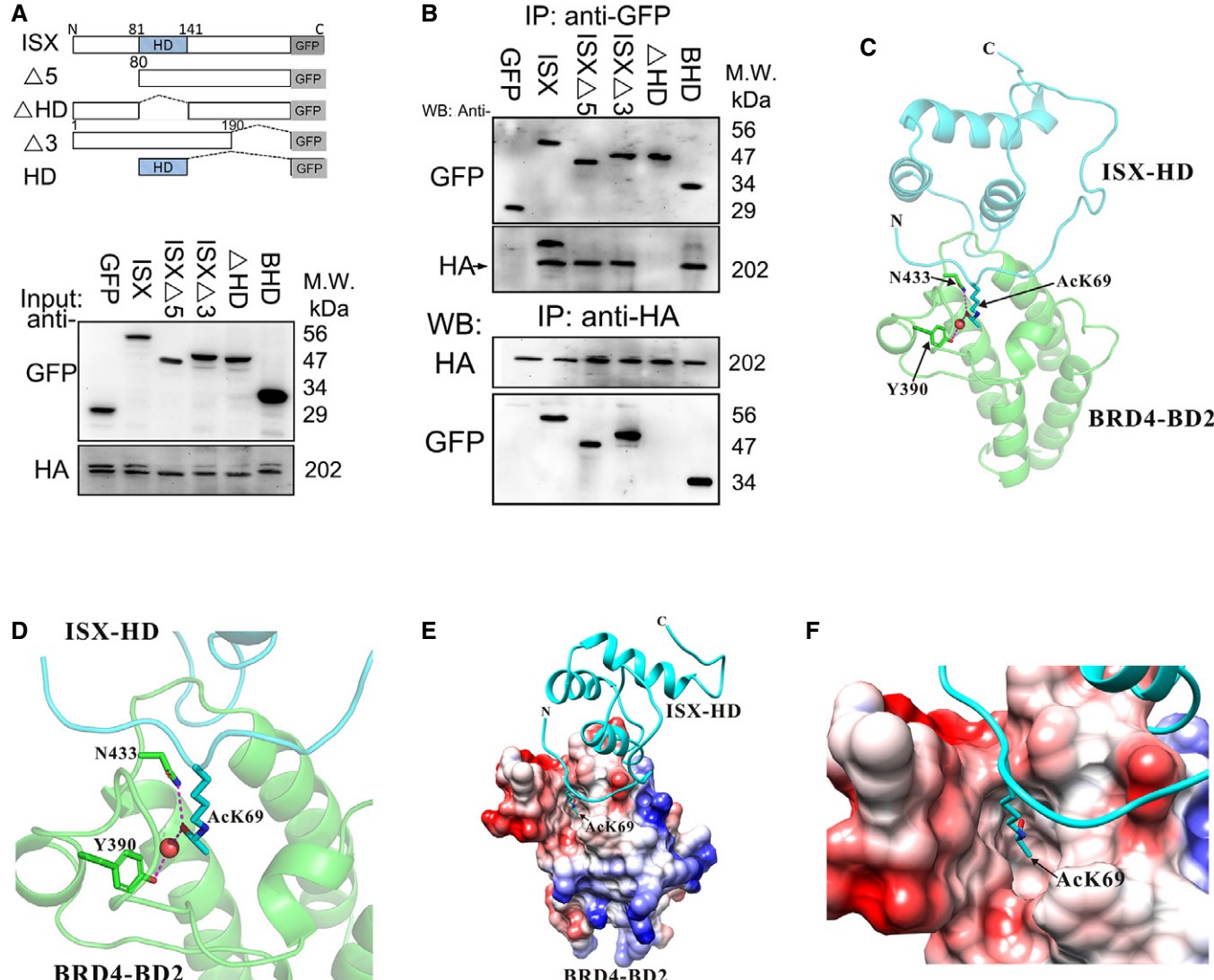

**Figure 5. Homeobox of ISX is a critical domain required for ISX–BRD4 interaction through co-immunoprecipitation (Co-IP) analysis.**

A Schematic depiction of ISX and deletion constructs used.
B GFP-tagged truncated ISX mutant proteins or HA-tagged BRD4 was detected in anti-HA or GFP immunoprecipitates by Western blotting analysis.
C–F A putative interaction surface of the ISX/BRD4 complex was presented through structural modeling. The electrostatic surfaces are drawn either blue for positive or red for negative charge (E and F).

Data information: Each experiment was repeated at least three times.
Source data are available online for this figure.

 

A549 cells abolished the mRNA enhancement of TWIST1 and Snail1 induced by forced ISX–BRD4 complex expression (Fig 4E and F), consequently abolishing its high DNA-binding affinity for the promoters of TWIST1 and Snail1 (Fig 4G and H). Moreover, A549 cells expressing the AC4 BRD4 mutant showed significantly decreased EMT characteristics (invasion activity) (Fig 4I).

### The homeobox of ISX is a critical domain that facilitates ISX–BRD4 complex formation via the BD2 domain of BRD4

To analyze the interaction mode of the ISX–BRD4 complex, co-immunoprecipitation was used to identify the interaction domain between ISX and BRD4 in A549 cells. GFP-tagged wild-type and deletion mutants of ISX [18] were transiently transfected into A549 cells with HA-tagged BRD4 expression (Fig 5A), after which immunoprecipitates by anti-GFP or anti-HA antibody were blotted (Fig 5B). HA-tagged BRD4 was detected in cells with wild-type and mutant ISX protein expression but not those with ISX ΔHD expression (upper panel in Fig 5B). Wild-type and mutant ISX, but not ISX ΔHD, also showed high binding affinity to HA-tagged BRD4 (lower panel in Fig 5B). Further, interactions between domains on BRD4 and ISX were monitored. Through three-dimensional structure modeling

(Fig 5C and D), surface electrostatic forces propensity, and solvation energy (Fig 5E and F), bromodomain 1 (BD1) and 2 (BD2) of BRD4 showed potential binding to the homeobox domain of acetylated ISX. Tyr97 (Y) and Asn140 (N) of the BD1 domain and Tyr390 (Y) and Asn433 (N) of the BD2 domain in BRD4 were critical residues needed for BRD4 to recognize and bind the acetylated lysine peptide on ISX. Wild-type and mutant BRD4 were then transfected into lung cancer cells with ISX-GFP (Fig 6A, left). Wild-type and mutant proteins of BRD4, including ΔBD1(ΔB1), 97$^{Y \to A}$, and 140$^{N \to A}$, were detected in immunoprecipitates obtained with anti-GFP. ΔBD2(ΔB2), 390$^{Y \to A}$, and 433$^{N \to A}$ mutants of BRD4 appeared to have no ISX-binding ability in A549 cells (Fig 6A, right). Moreover, A549 cells expressing BD1(ΔB1), ΔBD2(ΔB2), 390$^{Y \to A}$, and 433$^{N \to A}$ mutants together with ISX-GFP showed significantly decreased DNA-binding activity of Snail1 and TWIST1 promoters (Fig 6B and C). The Hprt1 promoter region (-190-+40 bp; red) was used as a negative control. Subsequently, the expression of ΔBD2(ΔB2), 390$^{Y \to A}$, and 433$^{N \to A}$ ISX mutants in A549 cells abolished the enhanced cell migration and invasion activity induced by the ISX–BRD4 complex (Fig 6D and E). The results suggest that interactions between acetylated Lys69 of ISX and Tyr390 and Asn433 in the BD2 domain of BRD4 play a critical role in ISX–BRD4 complex formation.

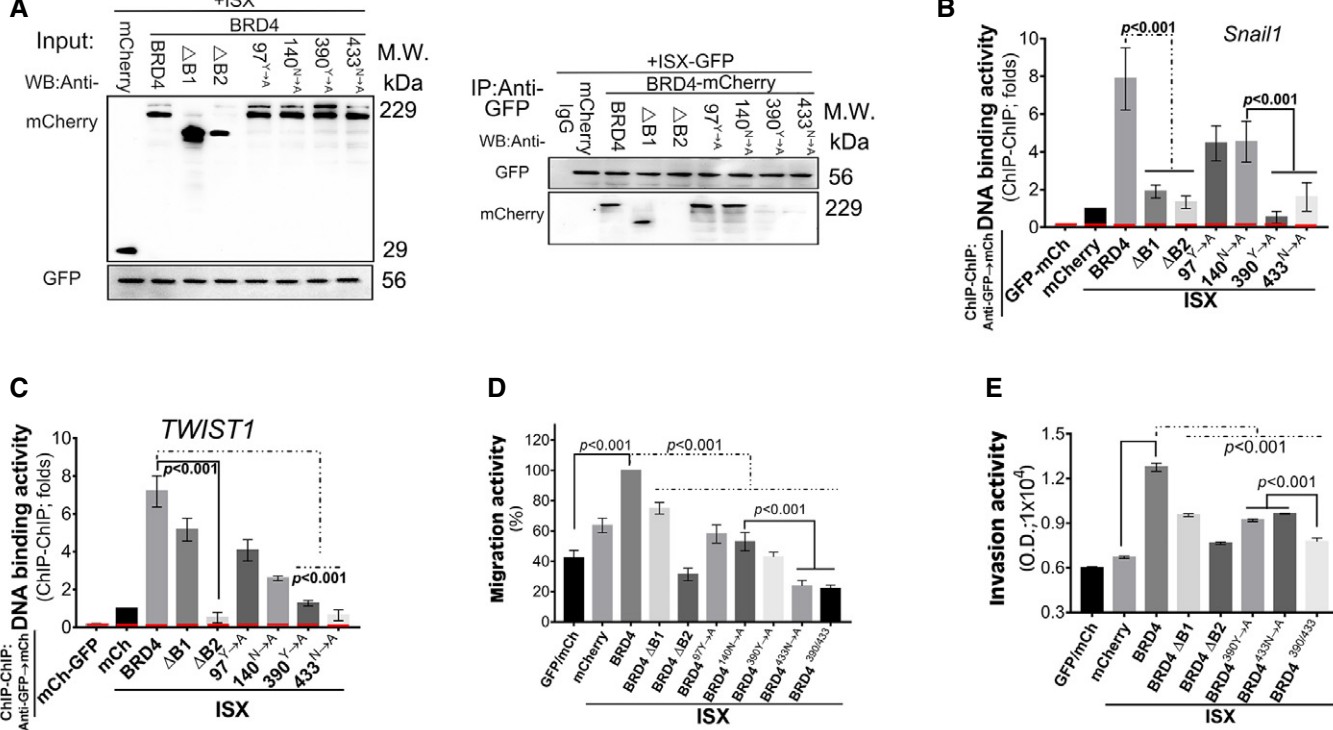

**Figure 6. BD2 domain of BRD4 is a critical domain that facilitates ISX–BRD4 complex formation.**

A mCherry-tagged truncated BRD4 mutant proteins and GFP-tagged ISX were detected in anti-GFP immunoprecipitates by Western blot.

B, C The DNA-binding activity of Snail (B) and TWIST1 (C) was evaluated in anti-GFP-mCherry ChIP–ChIP immunoprecipitates by RT–PCR in A549 cells. Red, Hprt1 promoter (−190 to +40 bp, negative control). (10% input of each group was pull down and applied to qPCR) Data are presented as mean ± SD in bar graph (P < 0.001, Student's t-test) of three independent experiments, each performed in triplicate.

D, E The cell migration (wound healing, D) and invasion (Transwell, E) activity were determined in A549 cells co-transfected with cDNAs for GFP-tagged ISX and mCherry-tagged BRD4 mutants. Data are presented as mean ± SD in bar graph (P < 0.001, Student's t-test) of three independent experiments, each performed in triplicate.

Data information: Each experiment was repeated at least three times.
Source data are available online for this figure.

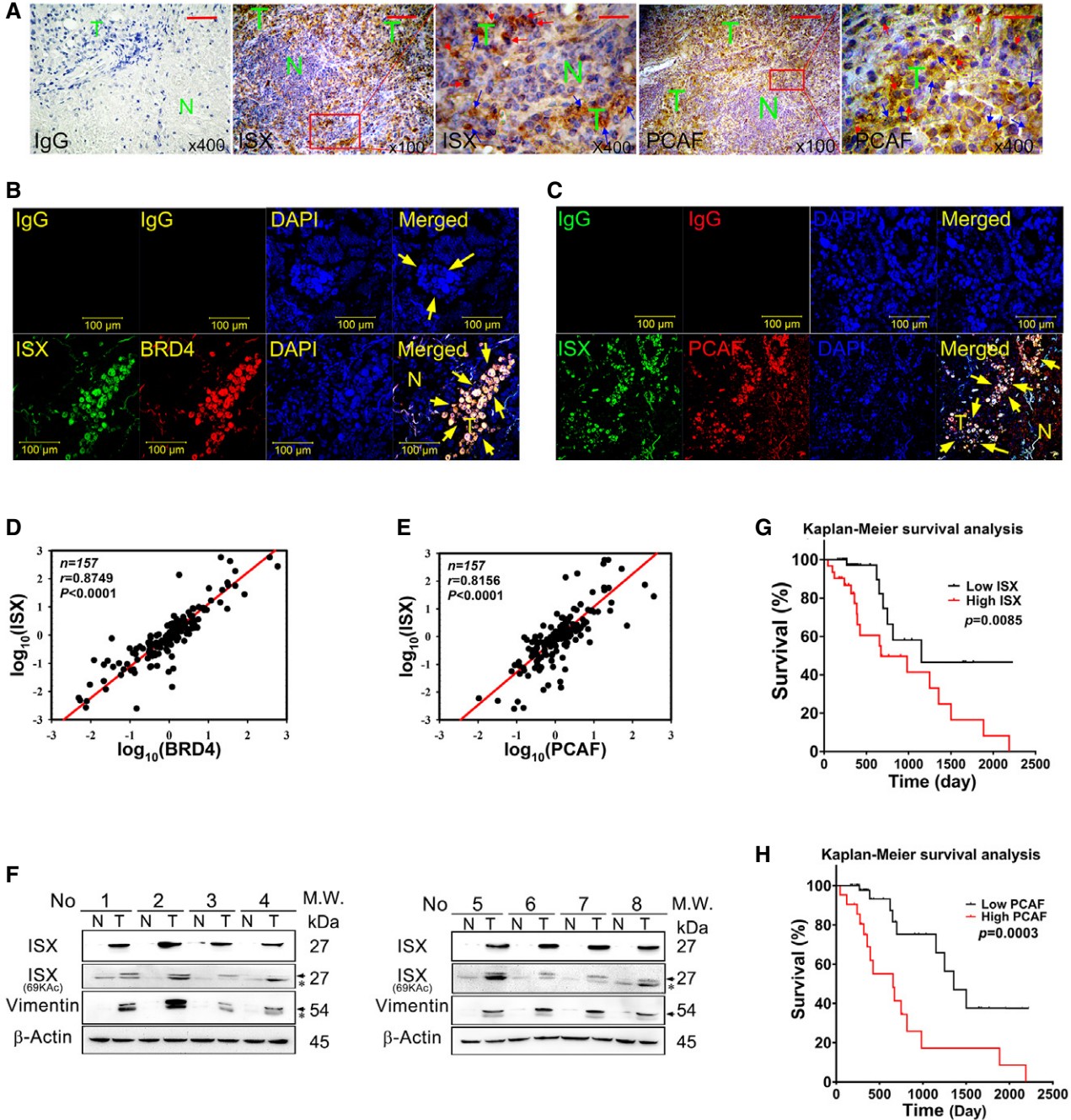

**Figure 7. The expression of ISX, BRD4, and PCAF is correlated with lung cancer metastasis.**

A   IHC staining with ISX (brown) or PCAF (brown) antibody in lung tumors from patients with lung cancer. N, normal tissue; T, tumor mass; Nuclei, hematoxylin (blue); Blue and red arrow, cytosol and nuclear ISX (PCAF). Scale bar, 500 μm (400×); Scale bar, 2 mm (100×).

B   Confocal immunofluorescence detection of ISX (green) and BRD4 (red) in lung tumors from patients with NSCLC. Cell nuclei were visualized by DAPI (blue); yellow arrows indicate co-localization; IgG, negative control. N, normal tissue; T, tumor mass.

C   Confocal immunofluorescence detection of ISX (green) and PCAF (red) in lung tumors from patients with NSCLC. Cell nuclei were visualized by DAPI (blue); yellow arrows indicate co-localization; IgG, negative control. N, normal tissue; T, tumor mass.

D, E   Correlation analysis of mRNA expression for, (D) ISX and BRD4, (E) ISX and PCAF in 157 lung cancer samples.

F   Protein expression of ISX, ISX(69Kac), and vimentin in tumor and normal parts from patients with NSCLC by Western blotting. N, normal tissue; T, tumor mass; Arrow, target protein; *, unspecific band.

G, H   The Kaplan–Meier survival curve was used to analyze survival correlation between patients ($n$ = 157) with NSCLC and ISX (G) and PCAF (H) levels. On the basis of the cut-off values of fold differences, the study population was dichotomized into the "high" and "low" expression groups. *P*-values were calculated by log-rank (Mantel–Cox) test comparing the two Kaplan–Meier curves.

Source data are available online for this figure.

## Expression levels of PCAF–ISX–BRD4 axis components show high correlation with clinical outcomes and prognosis in patients with NSCLC

To explore the clinical impact of PCAF–ISX–BRD4 signals in lung cancer, 157 paired NSCLC tumor samples (tumors along with neighboring healthy lung tissues) were obtained and analyzed. Compared with the adjacent normal lung tissues, both ISX (brown) and PCAF (brown) showed a tumor-specific expression pattern in lung tumor masses and were detected in both the cytoplasm and the nuclei of tumor cells (Fig 7A). Confocal fluorescence imaging was then used to examine the interaction between ISX, BRD4, and PCAF in immunostained samples of tumor masses and adjacent healthy lung tissues from patients with NSCLC. The expression of both BRD4 (red; Fig 7B) and PCAF (red; Fig 7C) was co-localized with ISX expression (green) in lung cancer cells (yellow arrow; Fig 7B and C). Moreover, mRNA expression of *ISX* strongly correlated with those of *BRD4* and *PCAF* in patients with NSCLC (Pearson's correlation coefficient, $r = 0.8749$ and $0.8156$, respectively, $P < 0.0001$; Fig 7D and E). *BRD4* mRNA expression also strongly correlated with *PCAF* mRNA expression in the same patients ($r = 0.8148$, $P < 0.0001$; Fig EV3D). Correlation of the expression for ISX, acetylated ISX, and invasion marker (vimentin) [20,21] was monitored in NSCLC tumor mass, and the results showed that acetylated ISX and total ISX expression correlated with the expression of vimentin and the level of invasion (Figs 7F and EV3E). To describe clinical characteristics and evaluate the prognostic value of ISX and BRD4 in NSCLC, 157 patients with follow-up data were analyzed for baseline characteristics. Patients with NSCLC were categorized into "low" and "high" expression groups for ISX and BRD4 using survival receiver operating characteristic curve analysis (Table 1). Significant differences in tumor sizes, metastases, and cancer stages were observed between patients with high and low levels of ISX or BRD4 expression (Table 1). Analysis of the survival curves indicated that patients with NSCLC having relatively lower ISX expression had a significantly longer survival time than that in patients with NSCLC having relatively higher expression after pulmonary resection (Fig 7G; $P = 0.0085$). Similarly, patients with NSCLC having relatively lower BRD4 or PCAF expression had a significantly longer survival time than that in patients with NSCLC having relatively higher expression after pulmonary resection (Figs 7H and EV3F). These results suggest that the PCAF–ISX–BRD4 axis is involved in NSCLC progression and patient survival.

## Discussion

The present study provides evidence that PCAF acetylation of the oncogenic transcription factor ISX recruits the acetylated BET member BRD4 to translocate the complex into the nucleus, promoting the formation of organized nucleosomes on the promoters of downstream target genes, whereas PCAF acetylation of histone H3 at

**Table 1. Clinical and pathological characteristics of 157 lung cancer patients according to mRNA expression of ISX and BRD4.**

| | | ISX | | | BRD4 | | |
|---|---|---|---|---|---|---|---|
| | **Total** | **Low** | **High** | **P** | **Low** | **High** | **P** |
| *N =* | 157 | 112 (%) | 45 (%) | | 108 (%) | 49 (%) | |
| Age (mean ± SD) | 60.9 ± 8.6 | 61.1 ± 8.5 | 60.6 ± 8.4 | 0.81 | 61.2 ± 8.0 | 60.6 ± 8.9 | 0.75 |
| Sex | | | | | | | |
| Male | 88 | 63 (56.2) | 25 (55.6) | 0.92 | 61 (56.5) | 27 (55.1) | 0.99 |
| Female | 69 | 49 (43.8) | 20 (44.4) | | 47 (43.5) | 22 (44.9) | |
| Size (cm) | | | | | | | |
| < 3 | 52 | 46 (41.1) | 6 (13.3) | < 0.01* | 44 (40.7) | 8 (16.3) | < 0.01* |
| 3≦ | 105 | 66 (58.9) | 39 (86.7) | | 64 (59.3) | 41 (83.7) | |
| LVI | | | | | | | |
| No | 83 | 63 (56.3) | 20 (44.4) | 0.24 | 61 (56.5) | 22 (44.9) | 0.24 |
| Yes | 74 | 49 (43.8) | 25 (55.6) | | 47 (43.5) | 27 (55.1) | |
| Meta | | | | | | | |
| No | 140 | 109 (97.3) | 31 (68.9) | < 0.01* | 105 (97.2) | 35 (71.4) | < 0.01* |
| Yes | 17 | 3 (2.7) | 14 (31.1) | | 3 (2.8) | 14 (28.6) | |
| Stage | | | | | | | |
| I | 71 | 60 (53.5) | 11 (24.4) | < 0.01* | 55 (50.9) | 16 (32.7) | < 0.01* |
| II | 32 | 19 (17.0) | 13 (28.9) | | 22 (20.4) | 10 (20.4) | |
| III | 37 | 30 (26.8) | 7 (15.6) | | 28 (25.9) | 9 (18.3) | |
| IV | 17 | 3 (2.7) | 14 (31.1) | | 3 (2.8) | 14 (28.6) | |

Lung cancer patients were classified into two groups—"low" and "high" according to survival receiver operator characteristic (ROC) curve analysis. The cutting points of ISX and BRD4 separately were 2.0 and 3.0 times of the mRNA expression in lung cancer tumors than that of the neighboring healthy tissues. LVI, lymphovascular invasion; SD, standard deviation. Statistical analysis of categorical variables was carried out by one-way ANOVA.
*$P < 0.05$.

lysine residues 9, 14, and 18 initiates chromatin remodeling and subsequent transcription (see Fig EV4 for a model). The PCAF–ISX–BRD4 axis upregulates EMT-associated gene expression and promotes tumor metastasis *in vitro* and *in vivo*. These results suggest that PCAF–ISX–BRD4 signaling plays a pivotal role in tumor metastasis and can be a potential therapeutic target for ISX-induced tumors.

Lung cancer is the leading cause of cancer deaths worldwide, including the United States, and metastasis is the major factor causing these deaths [22,23]. Metastasis is a complex multistep morphogenic process, and epithelial–mesenchymal transition (EMT) is believed to be the initial step of this process [24]. Despite several signaling molecules (e.g., PI3K, Snail, HIF1α, and SIP1) [25–28] and transcription factors (TWIST1/2, Snail1/2, ZEB1/2, and FOXC2) [29–32] having been identified as major regulators of EMT, a detailed regulatory mechanism for oncogene-induced EMT has yet to be established. The present study shows that the oncogenic transcription factor ISX is a key regulator of EMT and tumor metastasis by modulating the expression of EMT regulators, such as TWIST1 and Snail1. ISX, a pair-family homeobox TF, is a pro-inflammatory cytokine (IL-6)-induced homeobox gene that is highly expressed in hepatoma cells from patients with hepatocellular carcinoma (HCC). By directly regulating downstream cell cycle regulators (cyclin D1 and E2F1) [18,19] and immune checkpoint regulators (IDOs, PD-L1, and B7-2) [17], ISX has been shown to promote cell tumorigenic activities and is highly correlated with patient poor prognosis, highlighting its importance in regulating HCC progression [18]. The findings of the present study further interpret the mechanism and correlation between ISX and poor prognosis in lung cancers, providing a new therapeutic target in cancer therapy.

PCAF, a member of the GCN5-related N-acetyltransferase family of protein acetyltransferases, has been shown to be involved in the modulation of tumorigenesis and metastasis; however, the pathogenic function of PCAF appears to be controversial [5,6] and is believed to be cofactor-dependent in human disease [7,33]. Although the oncogenic function of PCAF had originally been identified to be induced by adenoviral E1A infection [5], it was also found to promote lung cancer progression by priming EZH2 acetylation [6]. Through acetylation of EZH2, PCAF enhances the protein stability in order to suppress target gene expression and promote lung cancer cell migration and invasion [6]. Conversely, PCAF has also been proposed to suppress cancer metastasis by restraining the activity of TP53 [34,35] and TFs Gli1 [36] and PTEN [37] through acetylation, thereby regulaying protein stability and inhibiting the EMT of cancer cells. Nonetheless, detailed regulatory mechanisms remain largely unknown. Our findings showed that PCAF acetylation of ISX is the key step to translocate the PCAF–ISX–BRD4 complex into nuclei and subsequent nucleosome formation on promoters of target genes. It provides vehicle-based evidence to interpret how PCAF is transported into nuclei to acetylate target histone proteins and chromatin remodeling.

The importance of BRD4 in tumorigenesis has been demonstrated by its ability to bind to acetylated histones and regulate transcription of oncogenic genes involved in several types of cancers [12]. Many small-molecule inhibitors, such as JQ1, were developed to disrupt protein–protein interactions between BRD4 and acetyl lysine, which effectively block cell proliferation and cytokine production in acute inflammation in cancers [38]. As an epigenetic reader, BRD4 recognizes acetylated H3 and H4 lysine residues via

its bromodomains and activates downstream gene expression [39,40]. BRD4 is widely distributed along the whole genome, and many TFs and chromatin remodeling proteins have been shown to interact with BRD4 and define its selectivity for target genes in tumorigenesis, such as TP53, c-JUN, C/EBPα and β, c-MYC, and TWIST1 [12,14–16]. Generally, TFs are initially shown to recruit acetyltransferases, such as p300/CBP, that subsequently promote the acetylation of other non-histone proteins in the enhancer nucleosomes of the target genes. Therefore, recruitment of BRD4 to active enhancers is likely mediated by its interaction with acetylated histones, as well as direct binding to enhancer-associated TFs [10,40]. Here, we provide evidence showing that ISX functions as an enhancer-binding factor in translocating the BRD4–PCAF complex to associate with histone H3-acetylated K27 on targeted nucleosomes, whereas PCAF acetylates histone H3 at 9, 14, and 18 lysine to initiate consequent chromatin remodeling. As shown in structure modeling, the carboxyl group of *N*-acetyl Lys69 of ISX could provide direct hydrogen bonding with the amide nitrogen of Asn433 in the BD2 domain of BRD4 and a water-mediated hydrogen bond interaction with the hydroxyl group of Tyr390 in the BD2 domain of BRD4 to facilitate ISX–BRD4 complex formation. Moreover, single point mutations, either Tyr390 or Asn433, when substituted with alanine, abolished ISX–BRD4 complex formation and led to subsequent gene repression of EMT regulators. Modulating the target gene expression of EMT regulators driven by PCAF–TF (ISX)–BRD4 clearly demonstrates a mechanistically synergistic regulatory effect on cancer metastasis.

Collectively, we have shown that the PCAF acetylation of the ISX–BRD4 complex unpacks chromatin and activates the expression of EMT regulators through acetylation of histone H3, subsequently promoting EMT and metastasis. Our findings highlight the important regulatory role of acetylation signaling in the PCAF–ISX–BRD4 axis to promote EMT initiation and regulation during tumor metastasis, highlighting its potential as a therapeutic target for prevention of EMT and metastasis.

## Materials and Methods

### Animals and cell culture

Male 8 weeks BALB/c nu/nu mice (*N* = 30; randomly divided into three groups) were obtained from the National Laboratory of Animal Breeding and Research Center (Taipei, Taiwan) and housed according to the protocols established by the Animal Center of the Kaohsiung Medical University (Kaohsiung, Taiwan). Human lung cancer cell lines (A549, H358, H441, H1299, H1435, and H1437), and human diploid lung fibroblast W138 were obtained from the American Type Culture Collection (ATCC; Manassas, VA, USA) and maintained according to their protocols. The study was conducted with approval (IACUC-104181) from the ethics committee of Kaohsiung Medical University.

### Plasmids, cell lines, and other materials

Full-length ISX cDNA was amplified from a human testis cDNA library (GIBCO/BRL, Cheshire, UK) using PCR. ISX cDNA and mutant ISX were each subcloned into the pEGFP/C1 vector

(Clontech) to express a GFP-tagged ISX. Full-length BRD4 and mutants were inserted into the FLAG (or mCherry)-tagged BRD4. The PLKO.1.puro or.neo vector was used as a backbone for shRNAi constructs targeting ISX (sequence, 5′-CAAACTTGCATCCCTGT GCTA-3′). The PLKO.1.puro or.neo vector was used as a backbone for shRNAi constructs targeting PCAF(sequence 4, 5′-GTTGGCTATA TCAAGGATTAT-3′; sequence 6, 5′-TGGCATGTCCATTAGCTATTT-3′; sequence 7, 5′-TTAATGGGATGTGAGCTAAAT-3′). WI38, A549, H358, H441, H1299, H1435, and H1437 cell lines were subcultured and maintained according to ATCC protocols. Transfection was performed using a lipofectamine transfection kit (GIBCO/BRL). Cell lines from both ATCC and BRC have been thoroughly tested and authenticated; morphology, karyotyping, and PCR-based approaches were used to confirm the identity of the original cell lines.

### Western blotting and immunohistochemical analysis

Western blotting staining and immunohistochemical (fluorescence) staining were performed as described previously [17,19]. The primary antibodies used in this study were Snail (1:1,000 dilution; MABE167; Merck, Darmstadt, Germany), E-cadherin (1:1,000 dilution; #3195S; Cell Signaling Technology), PCAF (1:1,000 dilution; #3378S; Cell Signaling Technology), vimentin (1:2,000 dilution; GTX100619; GeneTex), β-actin (1:10,000 dilution; #4967L; Cell Signaling Technology), HA (1:1,000 dilution; #3724S; Cell Signaling Technology), GFP (1:500 dilution; SC-9996; Santa Cruz Biotechnology), ISX (1:200 dilution; sc-398934; Santa Cruz Biotechnology), TWIST1 (1:200 dilution; ab49254; Abcam), BRD4 (1:1,000 dilution; #13440S; Cell Signaling Technology), acetylated lysine (1:500 dilution; #9441S Cell Signaling Technology), fibronectin (1:500 dilution; GTX112794; GeneTex), mCherry (1:500 dilution; GTX128508; GeneTex), Slug (1:1,000 dilution; GTX128796; GeneTex), VEGF (1:200 dilution; sc-7269; Santa Cruz Biotechnology), and N-cadherin (1:1,000 dilution; GTX127345; GeneTex). FITC-conjugated anti-rabbit IgG, rhodamine-conjugated anti-mouse IgG, and alkaline phosphatase-conjugated anti-rabbit IgG antibody (1:500 dilution; Jackson ImmunoResearch Laboratories, West Grove, PA, USA) were also used. All experiments were repeated at least three times.

### In vitro acetyltransferase assays

In vitro acetylation assay was performed according to published protocols with minor modifications [41]. Briefly, 20 ng of KAT2B (PCAF) and 100 ng of His6-ISX (BRD4 or their mutants) were combined in 40 μl reactions containing 50 mM Tris-Cl (pH 8.0), 5% glycerol, 2 mM DTT, 0.1 mM EDTA, and variable amounts of [$^3$H]-acetyl-CoA (30 nM). Reactions were performed for 30 min at 30°C and stopped by addition of an equal volume of 2× sample buffer followed by heating for 15 min at 100°C. A fraction of the final product was resolved by 10% SDS–PAGE and analyzed by Western blotting.

### Co-immunoprecipitation (IP and Co-IP)

Whole-cell lysates from $2 \times 10^7$ cells were prepared in modified RIPA (or RIPA) buffer (50 mM Tris-Cl (pH7.5), 150 mM NaCl, 1 mM EDTA, 1% NP-40 (0.2% SDS), and 0.5% Na-deoxycholate). The modified RIPA buffer was used to perform histone H3 extracts.

After centrifugation, the supernatant was incubated with 2 μg of antibodies as indicated, and then, protein A/G Sepharose bead was added and the incubation was continued at 4°C. The beads were washed three times with 1,000 μl of RIPA buffer and examined by Western blot analysis.

### Two-dimensional electrophoresis and in-gel digestion

Two-dimensional electrophoresis and In-gel digestion analyses were performed as described previously with modifications [42].

### Proximity ligation assay

Proximity ligation assay (PLA) was done using the Duolink® In Situ Red Starter Kit Mouse/Rabbit kit (Duolink Sigma, St. Louis, USA), following the manufacturer's instructions. Antibodies identifying ISX and BRD4 (PCAF) of human origin were used. Briefly, mouse monoclonal ISX (Santa Cruz Biotechnology) and rabbit polyclonal anti-BRD4 (PCAF) antibody (Cell Signaling Technology) were tagged with anti-mouse MINUS and anti-rabbit PLUS probes and applied on the cell. Following several washing and ligation steps as per the kit instructions, the signal was visualized using signal amplification of red fluorescent detection agent. The cells were mounted with DAPI and the coverslip sealed.

### Cytoplasmic and nuclear protein fraction preparation

Cytoplasmic and nuclear protein fractions were isolated from lung cancer cell lines using a nuclear extraction kit (ab113474; Abcam Biotechnology, Cambridge, United Kingdom) according to the manufacturer's instructions.

### Structural modeling

In order to interpret putative interactions between ISX and BRD4 from a structural point of view, we created a 3D structural model of the BRD4 BD2 domain (BRD4-BD2) in complex with the home-odomain of ISX (ISX-HD). The structural model of ISX-HD was generated by the HHpred server [43], a website that uses HMM–HMM comparisons to detect homology and predict structures, using the Protein Data Bank (PDB) code 2KT0 (the homeodomain of human Nanog) as the template. In the modeled structure of ISX-HD, Cα atoms between residues K69 to K72 were superimposed to Cα atoms between residues AcK5 to AcK8 in chain B of PDB code 3UVW, the crystal structure of human BRD4-BD1 in complex with a diacetylated histone H4 peptide (H4AcK5AcK8), using the Superimpose program of the CCP4 suite [44,45]. Subsequently, the coordination of K69 from the superimposed ISX-HD model was replaced by the coordination of AcK5 from the diacetylated histone H4 peptide of PDB code 3UVW to generate the AcK69 ISX-HD model. Cα atoms in chain A of PDB code 5UEZ, the crystal structure of BRD4-BD2, were superimposed with Cα atoms in chain A of PDB code 3UVW (BRD4-BD1). The coordination of the AcK69 ISX-HD model was merged with the coordination of the superimposed BRD4-BD2 model of 5UEZ to generate the BRD4-BD2/ISX-HD (AcK69) complex model, which was then energy-minimized using Swiss-PdbViewer [46]. The structural figures were then produced using PyMOL (DeLano Scientific; http://www.pymol.org) and UCSF Chimera [47].

## Luciferase reporter assays

The Snail1 and TWIST1 promoters were cloned respectively from human placenta genomic DNA and were used to construct a pGL3 luciferase reporter plasmid. The expression constructs and two reporter constructs, pSV40-Rluc and pGL3-Snail1 or TWIST1/Fire luciferase (Promega Co., Madison, WI, USA), were co-transfected with ISX into $2 \times 10^5$ A549 cells. The cells were harvested 16 h after transfection, and the relative luciferase activity was measured according to the manufacturer's instructions. All data are expressed as mean $\pm$ SD. of at least three experiments.

## Real-time PCR

The expression of *ISX, BRD4,* and *PCAF* mRNA in lung cancer cells and cells from cancer patients was quantified using an SYBR Green Quantitative RT–PCR kit (Invitrogen) as described previously. Total RNA was extracted from tumor mass using TRIzol reagent (Invitrogen) and then transcribed into cDNA (Invitrogen) for PCR amplification using a 7900HT Thermocycler (Thermo Fisher Scientific, Waltham, MA, USA). All procedures and data analysis were performed according to the manufacturers' instructions. The cells were transfected with an empty pEGFP vector, and samples from healthy subjects and drug-treated patients were analyzed for comparison. All data are expressed as mean $\pm$ SD of at least three experiments.

## Wound-healing assay

Human lung cancer cells overexpressing *ISX* or transfected with shRNA that were seeded in 10-cm culture plates were scratched using a pipette tip to create a gap, followed by incubation at 37°C and imaged every 12 h using a digital camera attached to a microscope. For each gap, the average width was computed from three measurements taken at the top, middle, and bottom end of the microscopic field.

## Transwell invasion assay

The Transwell invasion assay was performed using a Transwell chamber (Life Technologies) with a Matrigel-coated filter. Human lung cancer cells $(1 \times 10^5)$ overexpressing *ISX* or transfected with shRNA were added to 250 μl of serum-free media and plated onto the upper chamber of the Transwell. The upper chamber was then transferred to a well containing 500 μl of media supplemented with 10% FBS and incubated for 18 h. Cells may actively migrate from the upper to the lower side of the filter using FCS as an attractant. Cells on the upside were removed using cotton swabs, and the invasive cells on the lower side were fixed, stained with a 0.2% crystal violet solution, and counted under a light microscope. The experiment was repeated three times.

## Chromatin immunoprecipitation (ChIP) and two-step ChIP assays

ChIP and two-step ChIP assays were performed as described elsewhere [48]. Briefly, A549 cells were treated with 1.0% formaldehyde to crosslink proteins to DNA for 10 min at room temperature.

Cells were spin down and lysed by the RIPA lysis buffer. Genomic DNA was sheared by sonication for 10 min at the M2 intensity level to acquire optimal DNA fragment size of ~300 bp. Immunoprecipitation was achieved by the addition of anti-GFP (ISX) antibody or Normal Rabbit IgG to the samples with rotation at 4°C overnight. Protein A/G agarose beads were added to the samples and incubated for 1 h at 4°C with rotation. Beads were washed by 10 mM PBS and then eluted by 0.1 M glycine solution (pH2-3). The eluted proteins from first immunoprecipitation step were then precipitated by addition of anti-mCherry (BRD4) antibody to samples or Normal Rabbit IgG as negative control sample and followed by second round IP step. Next day, samples were washed by 10 mM PBS and then eluted by TE buffer for quantitative PCR. The TWIST1 (−186 to +35) and Snail1 (−160 to +40) promoter fragments were amplified with the following primers: TWIST1 Primer 1, 5′-GGACGAATTGT TAGACCCCG-3′; TWIST1 Primer 2, 5′-CCGGTGCTGCAGAGCCC GCG-3′; Snail1 Primer 1, 5′-GGCCAGCAGCCGGCGCACCT-3′; Snail1 Primer 2, 5′-GCGCAGAAGAACCACTCGCT-3′.

## Determination of tumor growth and metastasis by IVIS analysis

To perform a tumor growth and metastasis analysis *in vivo*, vector- or ISX mutant-transfected A549-RFP cells $(1 \times 10^6)$ were injected into lung tissues of 6–8-week-old male nude mice, and the images were monitored using IVIS each week. Mice were imaged every week in the prone position in an IVIS200 Imaging System (Caliper Life Sciences, Hopkinton, MA, USA). Data were acquired and analyzed using the Living Image software, version 4.2.

## Patients

This study enrolled 157 patients with lung cancer from July 2013 to July 2018 from two medical centers in Taiwan: E-DA Hospital (107 patients) and Taiwan Lung Cancer Network (50 patients). The study of human subjects was approved by the Institutional Review Board of Kaohsiung Medical University (KMUHIRB-E(I)-20180250; Kaohsiung, Taiwan).

## Statistical analysis

Quantitative variables are presented as mean $\pm$ SD. Significance of differences was determined using a two-sample *t*-test. Pearson's correlation analysis was used to examine the relationship between expression levels of ISX, BRD4, and PCAF. Statistical analysis of categorical variables was performed using chi-squared analysis, one-way analysis of variance, and Fisher's exact analysis. Differences with a $P < 0.05$ were considered significant.

**Expanded View** for this article is available online.

## Acknowledgements

We are grateful for the supporting from the Bio-Bank, Medical Research Department, E-DA Hospital, Taiwan. This work was supported in part by research grants KMUH 106-6R33, KMUH107-7R46, and KMUH107-7R34 from the Kaohsiung Medical University Hospital, Kaohsiung Medical University Research Center Grant (KMU-TC108A04-5) and MOST-105-2314-B-037-70-MY2, MOST-106-2314-B-037-090-MY2, MOST-107-2314-B-037-063-MY3, MOST-107-2314-B-037-028-MY3, MOST-108-2320-B-037-005, and MOST-108-2314-B-037-065-MY3

from the Ministry of Science and Technology, Taiwan (MOST), Kaohsiung Medical University (KMU) grant (KMU-DK108012; KMU-TC108A02), National Health Research Institutes (NHRI), Taiwan (EOPP10-014; NHRI-EX108-10720SI); Kaohsiung Medical University "The Talent Plan" (106KMUOR04), Taiwan; and Shenzhen Science and Technology Peacock Team Project (KQTD20170331145453160). C.-M.C. is supported by US National Institutes of Health (NIH RO1CA103867), Cancer Prevention & Research Institute of Texas (CPRIT RP180349 and RP190077), and Welch Foundation (I-1805).

## Author contributions

Conception and design: S-HH and L-TW Development of methodology: S-HH and L-TW. Acquisition of data (provided animals, acquired and managed patients, provided facilities, etc.): L-TW, C-YC, M-SH, and S-NW. Analysis and interpretation of data (e.g., statistical analysis, biostatistics, computational analysis): S-HH, L-TW, K-YL, W-YJ, and S-NW. Writing, review, and/or revision of the manuscript: S-HH, L-TW, C-MC, S-KH, and K-KY. Administrative, technical, or material support (i.e., reporting or organizing data, constructing databases): S-HH and S-SC Study supervision: S-HH and L-TW.

## Conflict of interest

The authors declare that they have no conflict of interest.

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
