## [Review Process File · EMBO Reports]

PCAF-mediated acetylation of ISX recruits BRD4 to promote epithelial-mesenchymal transition

Li-Ting Wang, Kwei-Yan Liu, Wen-Yih Jeng, Cheng-Ming Chiang, Chee-Yin Chai, Shyh-Shin Chiou, Ming-Shyang Huang, Kazunari K. Yokoyama, Shen-Nien Wang, Shau-Ku Huang and Shih-Hsien Hsu

Review timeline:

Submission date:	4 July 2019
Editorial Decision:	2 August 2019
Revision received:	19 September 2019
Editorial Decision:	6 November 2019
Revision received:	8 November 2019
Accepted:	22 November 2019

Editor: Achim Breiling

Transaction Report:

1st Editorial Decision

2 August 2019

Thank you for the submission of your research manuscript to EMBO reports. We have now received reports from the three referees that were asked to evaluate your study, which can be found at the end of this email.

As you will see, all referees think that the findings are of interest, but they also have several comments, concerns and suggestions, indicating that a major revision of the manuscript is necessary to allow publication in EMBO reports. As the reports are below, I will not detail them here. However, it will be of greatest importance to provide evidence that endogenous proteins indeed form a complex (PCAF-ISX-BRD4), and ChIP using endogenous DNA needs to be done. We do not think it would be necessary to raise antibodies against specific acetylated proteins (as suggested by referee #1), if an alternative strategy is used (e.g. IP and the Western blot with pan-ac antibodies).

Given the constructive referee comments, we would like to invite you to revise your manuscript with the understanding that all referee concerns must be addressed in the revised manuscript and in a detailed point-by-point response. Acceptance of your manuscript will depend on a positive outcome of a second round of review. It is EMBO reports policy to allow a single round of revision only and acceptance or rejection of the manuscript will therefore depend on the completeness of your responses included in the next, final version of the manuscript.

Revised manuscripts should be submitted within three months of a request for revision; they will otherwise be treated as new submissions. Please contact me if a 3-months time frame is not sufficient so that we can discuss the revisions further.

When submitting your revised manuscript, please also carefully review the instructions that follow below. Failure to include requested items will delay the evaluation of your revision. When submitting your revised manuscript, we will require:

1) a .docx formatted version of the final manuscript text (including legends for main figures, EV figures and tables), but without the figures included. Please make sure that the changes are highlighted to be clearly visible. Figure legends should be compiled at the end of the manuscript text.

2) individual production quality figure files as .eps, .tif, .jpg (one file per figure), of main figures and EV figures. Please upload these as separate, individual files upon re-submission.

The Expanded View format, which will be displayed in the main HTML of the paper in a collapsible format, has replaced the Supplementary information. You can submit up to 5 images as Expanded View. Please follow the nomenclature Figure EV1, Figure EV2 etc. The figure legend for these should be included in the main manuscript document file in a section called Expanded View Figure Legends after the main Figure Legends section. Additional Supplementary material should be supplied as a single pdf labeled Appendix. The Appendix should have page numbers and needs to include a table of content on the first page (with page numbers) and legends for all content. Please follow the nomenclature Appendix Figure Sx, Appendix Table Sx etc. throughout the text, and also label the figures and tables according to this nomenclature.

For more details please refer to our guide to authors:

See also our guide for figure preparation:

http://wol-prod-cdn.literatumonline.com/pb-assets/embosite/EMBOPress_Figure_Guidelines_061115-1561436025777.pdf

4) a complete author checklist, which you can download from our author guidelines (<https://www.embopress.org/page/journal/14693178/authorguide>). Please insert page numbers in the checklist to indicate where the requested information can be found in the manuscript. The completed author checklist will also be part of the RPF.

Please also follow our guidelines for the use of living organisms, and the respective reporting guidelines: <http://www.embopress.org/page/journal/14693178/authorguide#livingorganisms>

5) that all material and methods and references are included into the main manuscript file.

6) We strongly encourage the publication of original source data with the aim of making primary data more accessible and transparent to the reader. The source data will be published in a separate source data file online along with the accepted manuscript and will be linked to the relevant figure. If you would like to use this opportunity, please submit the source data (for example scans of entire gels or blots, data points of graphs in an excel sheet, additional images, etc.) of your key experiments together with the revised manuscript. If you want to provide source data, please include size markers for scans of entire gels, label the scans with figure and panel number, and send one PDF file per figure.

7) Our journal encourages inclusion of *data citations in the reference list* to directly cite datasets that were re-used and obtained from public databases. Data citations in the article text are distinct from normal bibliographical citations and should directly link to the database records from which the data can be accessed. In the main text, data citations are formatted as follows: "Data ref: Smith et al, 2001" or "Data ref: NCBI Sequence Read Archive PRJNA342805, 2017". In the Reference list, data citations must be labeled with "[DATASET]". A data reference must provide the database name, accession number/identifiers and a resolvable link to the landing page from which the data can be accessed at the end of the reference. Further instructions are available at:

8) Regarding data quantification and statistics, can you please specify, where applicable, the number "n" for how many independent experiments (biological replicates) were performed, the bars and error bars (e.g. SEM, SD) and the test used to calculate p-values in the respective figure legends. Please provide statistical testing where applicable, and also add a paragraph detailing this to the methods section. See:

<http://www.embopress.org/page/journal/14693178/authorguide#statisticalanalysis>

Finally, please note that all corresponding authors are required to supply an ORCID ID for their name upon submission of a revised manuscript. Please find instructions on how to link ORCID IDs to the accounts in our manuscript tracking system in our Author guidelines:

<http://www.embopress.org/page/journal/14693178/authorguide#authorshipguidelines>

I look forward to seeing a revised version of your manuscript when it is ready. Please let me know if you have questions or comments regarding the revision.

REFEREE REPORTS

Referee #1:

In this report, Wang et al. described a story that PCAF Acetylation of ISX Recruits BRD4 to Promote Epithelial-Mesenchymal Transition. They showed that a PCAF-ISX-BRD4 axis mediates EMT signaling and exerts regulatory effects on tumor initiation and metastasis. Their findings are potentially interesting. However, many holes in this study prevent its publication. Concerns are shown below:

1. In general, the figure legends are poorly written and ambiguous.
2. Fig. 2. e and f are wrongly labeled.
3. In the localization picture, what arrows refer to must be described.
4. Fig. 3C, I would like to see the endogenous interaction of PCAF with ISX and BRD4.
5. In Fig. 3 d and e I would like to see the sequential co-IP to confirm that PCAF with ISX and BRD4 could complex together. Now the evidence for them to form a complex is very weak.
6. The big problem for the present investigation is the lack of evidence that acetylation of ISX and BRD4 do exist in living cells. I suggest the authors to make BRD4-K-332 and ISX-K-69 site-specific acetylation antibodies. These antibodies are very important for confirming the acetylation status of ISX and BRD4 in cells. No one knows if BRD4-K-332 and ISX-K-69 are the major acetylation sites in living cells until the site-specific acetylation antibodies are used.
7. For Figure 5. a and b, the authors must provide with detailed protocol for ChIP-ChIP assay.
8. It is weird and not necessary, abbreviations "ISX, intestine-specific homeobox; BRD4, bromodomain-containing protein 4; PCAF, P300/CBP-associated factor" were shown in the figure legends 2, 3, 5, 6 and 7.
9. Figure 7, what the authors were trying to show is not consistent with the purpose of this investigation. What we concern is the acetylation degree of ISX and BRD4 in the patients and whether the level of ISX and BRD4 acetylation correlate with the progression of lung cancer patients.

Referee #2:

Hsu and colleagues describe a novel epigenetic mechanism of regulation of EMT and metastasis during tumor progression of lung cancer. On a molecular level the authors dissected that ISX is interacting with PCAF and BRD4 to form a complex that is able to enter the nucleus and activate expression of the EMT transcription factors Twist and Snail in order to promote EMT features in A549 cells. Experimental manipulation of ISX expression resulted in acquisition of an EMT phenotype, combined with increased migration and invasion and facilitation of tumor growth and metastasis upon orthotopic transplantation. The authors identified K69 in ISX to be acetylated by PCAF to induce recruitment of BRD4 for nuclear translocation and chromatin remodeling, i.e. H3 acetylation and gene activation. Furthermore the authors identified the binding interface between

ISX and BRD4 and also correlated elevated expression of the three factors with poor prognosis. These data are of particular interest since the mechanism about how EMT is initiated upstream of EMT-TFs is still not completely understood. Therefore, the manuscript merits publication in EMBO reports. However, I raised some important technical and conceptual points that need to be addressed before publication. Although the analysis is properly structured and experiments are providing very valuable data to support the conclusions, the manuscript shows some important caveats and several experiments lack a proper description and/or important controls. For some experiments the description is either misleading or incorrect.

I listed my concerns below.

1. Luciferase reporter analysis, Fig. 1c+d: It is not clear to me how the data have been generated. It looks like that in all samples ISX was cotransfected and values of pGL3basic were set to 1. It would be important to see promoter activities with AND without ISX ectopic expression in parallel. Otherwise ISX specific effects cannot be discriminated from basal promoter activities. The pGL3basic control should also be shown as first column in 1d. More importantly, the short identified DNA sequences in both Twist and Snail promoters which are required for ISX-mediated transactivation are not characterized further. Do they contain common TF binding motifs? Do the results indicate that another unknown transcription factor is necessary to recruit PCAF-ISX-BRD4 to the specific promoters?

2. ChIP experiments in Figs. 1, 5a-b and 6b-e: In Fig. 1 the precipitated DNA of the plasmids were used to identify binding to the promoter regions (which primers were used?). These experiments are only giving hints of binding, but are not very conclusive. Rather ChIP on endogenous chromatin needs to be done with endogenously expressed ISX protein (maybe in combination with overexpression and ISX knockdown). Enrichment needs to be presented as % of input. I am puzzled what is shown in Fig. 5a-b? Is this a reChIP experiment as indicated by the axis label 'ChIP-ChIP'? Which antibodies were used? Which genomic region was analyzed? Were endogenous Twist/Snail loci or again luciferase plasmids precipitated? Which primer sequences were used? Based on the analysis in Fig. 1 overexpressed ISX alone should already bind to the Twist and Snail promoters, which is not visible in Fig. 5. Does Fig. 5 now show increased binding by BRD4 on top of the ISX-mediated binding of the complex? In order to solve this question mock-transfected controls should be included, as well as the analysis of a non-related chromatin region, e.g. Hprt1 or regions that are >20 kb up or downstream of the Twist and Snail loci. The same holds true for Fig. 6b+c!

3. Fig. 3f: These results are very puzzling. In this experiment a genome-wide anti-H3 IP was performed to check on the acetylation state of the protein. According to the text and the figure the authors claim that in control cells there is absolutely no acetylation of H3. Only upon forced ISX expression acetylation is detected. And this results in global changes of H3-Ac of the whole genome? How can this be explained, since there must be steady-state open chromatin regions in non-transfected control cells, marked by acetylated H3.

4. I have a conceptual problem with the orthotopic transplantation experiment in Fig. 4a: The problem with this approach is that tumor cells are not labeled independent of ISX labeling, e.g. by CMV/CAGGS-Luciferase or RFP. Hence, different expression levels of RFP and ISX-RFP might result in different BLI intensities and may be inadequate to monitor tumor cell numbers. Analysis of cryosections of lungs with RFP fluorescence detection would help to clarify this. If not available, at least tumor cells should be analyzed in vitro for RFP fluorescence intensities and heterogeneity by microscopy. Moreover, histological examination/quantification of tumor mass should be done to confirm the results from BLI. Do the tumors also display differences in EMT states? The results in Fig. 4a show that ISX K69 is responsible for efficient tumor growth. However, it cannot be concluded yet whether EMT or metastasis are affected as well.

5. Fig. 4 g+h, mRNA analysis: The results are confusing as they are inconsistent with findings in Fig. 1a. In Fig. 1 a specific effect on Twist and Snail mRNA by ISX ectopic expression alone was observed. Is the observed increase in Fig. 4 due to the combination of ISX/BRD4 on top of the effects of ISX alone? It would be helpful to include mock-transfected samples as well to clarify this. The Figs S2a and 4g+h should be consistent with Fig. 1, showing both Twist and Snail mRNA and protein for the ISX and BRD4-specific experiments.

6. P. 8, Fig. 2d: ISX Co-IP experiment: do the authors claim that in normal lung tissue there is LESS interaction/recruitment found or is there just simply less expression of CBP/CREB, PCAF present to result in weaker Co-IPs of the interaction partners? A specific reduction in interaction can only be claimed for BRD4, since it shows similar expression levels in normal vs. NSCLC, but less interaction to ISX. However, this might be related to the fact that other complex components are less abundant to form a stable ternary/quarternary complex!

7. Co-IP of the protein complex in Fig. 2c: all analyzed proteins bind with more or less the same extend to each other. Can it be excluded that they are indirectly linked via DNA fragments? The authors should provide another transcription factor/DNA associated protein as negative control which does not bind to most of these IP'd candidates.

8. IF in Fig. 2e+g: BRD4 supposed to be constitutively nuclear, also in A549 cells. The provided IF is not convincing and the discrepancy needs to be discussed. The sites of colocalization/dots are too small and a higher magnification should be included. To verify interaction of the proteins in situ, I suggest to do a Proximity Ligation Assay.

9. Fig. 4f, ISX-BRD4 co-IP: What is shown in Fig 4f in the blot with the label 'mCherry (Kac)'? Was a lysine acetylation-specific antibody used here? It would be helpful to also show the full blot of the IP including the Mw range of mCherry alone to demonstrate no acetylation of mCherry. This also applies to Fig. 4e (GFP).

10. The figure subdivision and distribution needs to be improved. Right now it is a bit random, with one experiment split into two main figures, e.g. Fig. 3i and 4a, Fig. 5a-c belong to Fig. 4.

11. Materials and methods miss important experimental procedures, e.g. how the Dox-inducible ISX expressing cell lines have been generated, which antibodies/primer sequences have been used, how ChIP experiments have been performed, etc.

Minor points:

1. P. 6: By describing the ChIP experiments in Fig. 1g,h the authors state that the 'aforementioned region showed no or lesser promoter activity than that with GFP expression control'. How can the DNA binding affinity report on the activity of the promoter and what is the GFP control in this experiment?
2. P.6. The 'IBE' abbreviation is used before it was defined one sentence later.
3. Frequently, the authors use the term 'significant' though no quantification or statistical analyses have been conducted, e.g. in Western blots of S1b on p.9 or Fig. 4e, p. 12.
4. Molecular weight markers are lacking in all Western blots. This is of particular importance for the IPs that show GFP-fusions vs. GFP-tag only in comparison.
5. Fig. 3g axis label: How is 'D/T' in terms of migration activity defined? Why is migration not displayed as in Fig. 2?

Referee #3:

The study submitted by Wang et al. identifies PCAF/ISX/BRD4 complex as an activator of Epithelial to Mesenchymal Transition. The authors highlight new targets of PCAF to induce EMT : PCAF acetylates ISX and BDR4 to induce a nuclear translocation of the complex that binds to the promoters of EMT regulators to activate transcription. Overall, data presented is convincing but figures legends are confusing, incomplete and need to be modified. The typography of the figures needs also to be streamlined. All details of the protein interactions are well described. The acetylation of ISX and BRD4 by PCAF appears crucial for EMT induction in NSCLC cellular models and for metastasis formation in mice. The PCAF-ISX-BDR4 axis seems to be also correlated to patient with NSCLC prognosis.

EMT study is a very competitive field of first interest in cancer biology. The study is interesting and follows along a similar line of previous papers. As mentioned in the study, the role of PCAF and BDR4 have already been shown in EMT. In this way, the authors might have referenced a recent

study published in CDD showing the involvement of BDR4 in SNAIL regulation at a transcriptional and protein stability levels in breast cancer (Cell Death Differ. 2019, doi: 10.1038/s41418-019-0353-2). The major novelty presented in this study is the implication of ISX in the PCAF/BDR4 translocation in NSCLC cancer to induce EMT.

Based on that some points that need to be addressed or discussed before considering the publication.

Major comments :

How many replicates are performed in western blotting experiments presented Figure 1b : (same remarks for all WB and IP experiments)?

I do not understand why the panel Figure 2a is not presented in the Figure 1. Invasion assays are complementary to migration assays and are not in adequacy with the title of the Figure 2 (« Effect of PCAF inhibitor on ISX-BRD4 complex formation, wound healing, and invasive activity »).

The immunofluorescence presented in Figure 2e and f are not convincing. Staining of ISX, BRD4 and PCAF are diffuse in the cells forcing the co-localizations. Proximity Ligation Assay experiments would be more obvious.

You present the expression profile of ISX, PCAF and BRD4 in supplemental data, the expression of these proteins seems to be correlated in NSCLC by contrast with WI38. What conclusion do you draw? Is there a common mode of regulation of these proteins/ARNm?

You used different acetyl transferase inhibitors : Tip60 inhibitor, PCAF, P300 inhibitors, GCN5 and you obtained the same effects in EMT inhibition when ISX is overexpressed. How you claim (in the title of this part) that PCAF modulates EMT induction? According to you this phenomenon is specific of PCAF or it exists a redundant function with other acetyltransferases? The specificity of these types of compounds can be discussed, can you use specific siRNAs?

Why do you perform an IP H3 (Figure 3f) to see the level of histones acetylation ? A western blotting after acid extraction of histones or an ELISA directed against these marks would have enough?

I do not understand why the panel a and b of the Figure 4 is not presented in the Figure 3. These two panels are showing effects of the absence of acetylation of ISX in tumor grow and are complementary to the panel i of the Figure 3.

The results obtained Figure 4a show the importance of ISX acetylation in tumor metastasis formation, can you validate this acetylation in your cohort of human tissues ?

Figure 7 b you cannot speak about « interaction » between ISX, BRD4 and PCAF based on the IF study but just co-localization. In the same way, you never demonstrate in my opinion, the direct interaction between the 3 proteins, or at least a complex formation including these 3 partners. To demonstrate the complex formation, you can perform a native gel for example with 1, 2 or the 3 proteins in the same sample or pull down experiments with these three purified proteins.

You clustered your NSCLC patients in Low and High expression of ISX. Can you determine the EMT status of your tissues using Vimentin staining as performed in these two studies (Vimentin expression predicts the occurrence of metastases in non-small cell lung carcinomas., doi: 10.1016/j.lungcan.2013.03.011 and EMT is associated with an epigenetic signature of ECM remodeling genes, doi: 10.1038/s41419-019-1397-4). This experiment could highlight a real correlation between your protein complex and EMT in patients with NSCLC.

Your final schematic diagram (Fig S3) needs to be modified: The representation of histones acetylation with the elimination of acetylated histone core from the DNA is naive (acetylation of histones leads just to chromatin decompaction). You detailed finely the acetylation sites of your proteins of interest by PCAF in the paper, can you add some details in your scheme (acetylated position...). The global design of your scheme needs to be refined.

Minor comments :

The abbreviation IBE needs to be defined before when you are presenting the IBE-containing region.

ChIP experiments (legend 1) are not described in Mat & Met as mentioned.

Co-IP-2D analysis is not presented in the Mat & Met.

Figure 1g-h : These panels can be placed in supplemental data (not essential, complementary to the panels c and d).

Figure 1c-d : Homogenize the presentation. Place -200 to +35 at the bottom, before d75-to -89 (as -90 to +40bp (d87-71bp) panel d).

Figure 1i : Explain ISXi in the legend.

Figure 2 legend: Points f and e : There is a mistake in the figure legend, e and f need to be changed.

Figure 2c legend: I do not understand why abbreviations of ISX, BRD4, PCAF, P30/CBP, EMT appear at the end of the legend.

Figure 2f comment « the expression of ISX and most BRD4 showed a significant cytosol-restricted pattern » replace expression by localization.

Supplemental data 1 : There is an empty box before the first amino acid, what is its role?

You need to mention TH1834 as a Tip60 inhibitor in the text (as mentioned for the other compounds)

Figure 2i-j : Streamline the presentation and add GFP after ISX if its presents: GFP/ vehicle, ISX/vehicle and ISX/Garcicol panel i and GFP/V, ISX-GFP/V and ISX/Garnicol. In the same way, for all protein overexpressed mentioned the name of the tag after the name of the protein of interest (GFP, HA, Cherry...) for a better understanding (for example Figure 2i-j, 3g-h...).

Figure 3 a : why do you mentioned the schematic representation of EZH2 (H and M) ? It is not indicated in the legend and on the core of the paper (same comment for Figure 4c).

Figure 3f : The sentence: « ISX interaction with histone H3 », is not correct. You are only showing the acetylation level of histones after overexpression of ISX WT and mutants.

Figure 4a : B and A at the right of the panel need to be eliminated. Streamline your groups (RFP, ISX-RFP, ISX Ac-RFP or ISX Ac3-RFP).

Figure 4f : Why AC3 construct is not presented in the figure?

1st Revision - authors' response

19 September 2019

Referee #1:

In this report, Wang et al. described a story that PCAF Acetylation of ISX Recruits BRD4 to Promote Epithelial-Mesenchymal Transition. They showed that a PCAF-ISX-BRD4 axis mediates EMT signaling and exerts regulatory effects on tumor initiation and metastasis. Their findings are potentially interesting. However, many holes in this study prevent its publication. Concerns are shown below:

1. *In general, the figure legends are poorly written and ambiguous.*

Response: We thank for the reviewer's comments. We have extensively revised the legends to improve their clarity; please see pages 35-41.

2. *Fig.2. e and f are wrongly labeled.*

Response: We thank for the reviewer's comment and have made the correction.

3. *In the localization picture, what arrows refer to must be described.*

Response: We thank for the reviewer's comments. We have added the description in the figure legends.

4. *Fig.3C, I would like to see the endogenous interaction of PCAF with ISX and BRD4.*

Response: We thank for the reviewer's comments. In the original manuscript, we provided data demonstrating the endogenous interaction of PCAF-ISX-BRD4 complex by co-IP in NSCL cells (Fig.2c) and tumor samples (Fig.2c), in conjunction with the use of confocal imaging (Fig.2e). In the revision, we have included new confocal images (Fig.S1f) and new data showing the results of proximity ligation assays (PLAs) (Figs.2d and g) of the PCAF-ISX-BRD4 complex, the results of which strongly support the existence of the endogenous complex.

5. *In Fig.3 d and e I would like to see the sequential co-IP to confirm that PCAF with ISW and BRD4 could complex together. Now the evidence for them to form a complex is very weak.*

Response: Please see the response above (Comment 4).

6. *The big problem for the present investigation is the lack of evidence that acetylation of ISX and BRD4 do exist in living cells. I suggest the authors to make BRD4-K-332 and ISX-K-69 site-specific acetylation antibodies. These antibodies are very important for confirming the acetylation status of ISX and BRD4 in cells. No one knows if BRD4-K-332 and ISX-K-69 are the major acetylation sites in living cells until the site-specific acetylation antibodies are used.*

Response: We thank for the reviewer's comments. In the original manuscript, both of the endogenous acetylated ISX-K69 and BRD4-K332 were detected by the use of anti-pan acetyl lysine antibody (Figs.3c, 3e and 4d) and LC-MS/MS analysis of the immune precipitates (Figs.S2b and S3b). As commented by the Reviewer, we have generated specific antibodies and the results (Fig. S1e) showed that the antibody generated against acetylated ISX-K69 peptide detected GFP-tagged wild-type ISX, but not GFP-tagged ISX-K69 mutant. The antibodies also detected acetylated ISX-K69 protein in NSCLC patient sample in which ISX-K69 protein level showed high correlation with EMT marker, Vimentin (Fig.7f). However, the antibody for acetylated BRD4-K332 generated against acetylated BRD4-K332 peptide failed to recognize endogenous acetylated BRD4-K332.

7. *For Figure 5. a and b, the authors must provide with detailed protocol for ChIP-ChIP assay.*

Response: We thank for the reviewer's comments. We described the detailed procedures for ChIP-ChIP assay in the Materials & Methods section as suggested by the reviewer.

8. *It is weird and not necessary, abbreviations "ISX, intestine-specific homeobox; BRD4, bromodomain-containing protein 4; PCAF, P300/CBP-associated factor" were shown in the figure legends 2, 3, 5, 6 and 7.*

Response: We thank for the reviewer's comments. We have made revisions of the respective figure legends. (please see pages 35-41).

9. *Figure 7, what the authors were trying to show is not concert with the purpose of this investigation. What we concern is the acetylation degree of ISX and BRD4 in the patients and whether the level of ISX and BRD4 acetylation correlate with the progression of lung cancer patients.*

Response: We thank for the reviewer's comments. To address this concern, we have generated anti-acetylated ISX-K69 Abs, and in eight paired NSLC lung tumor samples (Fig.7f), the relative levels of acetylated ISX-K69 were shown to be correlated with the level of ISX expression.

Referee #2:

Hsu and colleagues describe a novel epigenetic mechanism of regulation of EMT and metastasis during tumor progression of lung cancer. On a molecular level the authors dissected that ISX is interacting with PCAF and BRD4 to form a complex that is able to enter the nucleus and activate expression of the EMT transcription factors Twist and Snail in order to promote EMT features in A549 cells. Experimental manipulation of ISX expression resulted in acquisition of an EMT phenotype, combined with increased migration and invasion and facilitation of tumor growth and metastasis upon orthotopic transplantation. The authors identified K69 in ISX to be acetylated by PCAF to induce recruitment of BRD4 for nuclear translocation and chromatin remodeling, i.e. H3 acetylation and gene activation. Furthermore the authors identified the binding interface between ISX and BRD4 and also correlated elevated expression of the three factors with poor prognosis. These data are of particular interest since the mechanism about how EMT is initiated upstream of EMT-TFs is still not completely understood. Therefore, the manuscript merits publication in EMBO reports. However, I raised some important technical and conceptual points that need to be addressed before publication. Although the analysis is properly structured and experiments are providing very valuable data to support the conclusions, the manuscript shows some important caveats and several experiments lack a proper description and/or important controls. For some experiments the description is either misleading or incorrect. I listed my concerns below.

1. *Luciferase reporter analysis, Fig. 1c+d: It is not clear to me how the data have been generated. It looks like that in all samples ISX was cotransfected and values of pGL3basic were set to 1. It would be important to see promoter activities with AND without ISX ectopic expression in parallel. Otherwise ISX specific effects cannot be discriminated from basal promoter activities. The pGL3basic control should also be shown as first column in 1d. More importantly, the short identified DNA sequences in both Twist and Snail promoters which are required for ISX-mediated transactivation are not characterized further. Do they contain common TF binding motifs? Do the results indicate that another unknown transcription factor is necessary to recruit PCAF-ISX-BRD4 to the specific promoters?*

Response: We thank for the reviewer's comments and, indeed, this is an important point of concern. Firstly, the levels of luciferase activities were divided by the levels of DOX-induced GFP expression. Secondly, in the Figs. 1c and d, the luciferase activities driven by IBE truncated promoters were abrogated to pGL3 basic vector only, which demonstrated ISX regulated both TWIST1 and Snail1 expression by the IBE sequence in cells with ISX expression. We had moved pGL3basic control to first column in 1d. As in previous analysis, the IBE-containing short fragment did not contain other common TF binding motifs.

2. *ChIP experiments in Figs. 1, 5a-b and 6b-e: In Fig. 1 the precipitated DNA of the plasmids were used to identify binding to the promoter regions (which primers were used?). These experiments are only giving hints of binding, but are not very conclusive. Rather ChIP on endogenous chromatin needs to be done with endogenously expressed ISX protein (maybe in combination with overexpression and ISX knockdown). Enrichment needs to be presented as % of input. I am puzzled what is shown in Fig. 5a-b? Is this a reChIP experiment as indicated by the axis label 'ChIP-ChIP'? Which antibodies were used? Which genomic region was analyzed? Were endogenous Twist/Snail loci or again luciferase plasmids precipitated? Which primer sequences were used? Based on the analysis in Fig. 1 overexpressed ISX alone should already bind to the Twist and Snail promoters, which is not visible in Fig. 5. Does Fig. 5 now show increased binding by BRD4 on top of the ISX-mediated binding of the complex? In order to solve this question mock-transfected controls should be included, as well as the analysis of a non-related chromatin region, e.g. Hprt1 or regions that are >-20 kb up or downstream of the Twist and Snail loci. The same holds true for Fig. 6b+c!*

Response: We thank for the reviewer's comments. We have added the relevant information on ChIP and ChIP-ChIP in the Materials and Methods. Also, we have added description in the figure legend.

In Figure 5, in order to verify the effects of BRD4 on the expression of ISX-induced EMT markers, TWIST1 and Snail 1, NSLC cells with ISX expression were transfected with mock (vector only; V-

ISX), wild-type and BRD4 mutant (BRD4mutant-ISX), and the value of V-ISX was set as 1. As the Reviewer suggested, we have added another mock control group transfected with reporter constructs expressing GFP and mCherry only and the control of Hprt1 promoter in the revised manuscript.

3. *Fig. 3f: These results are very puzzling. In this experiment a genome-wide anti-H3 IP was performed to check on the acetylation state of the protein. According to the text and the figure the authors claim that In control cells there is absolutely no acetylation of H3. Only upon forced ISX expression acetylation is detected. And this results in global changes of H3-Ac of the whole genome? How can this be explained, since there must be steady-state open chromatin regions in non-transfected control cells, marked by acetylated H3.*

Response: We thank for the reviewer's comments. In Fig. 3f, decompacted histone H3 (free form and hyper acetylated H3) were pulled down for immune blotting, but not genome-wide H3. Different levels of H3 proteins were pulled down by cell lysis buffer (contained different detergent species and concentration) and sonicated the sample or not. In this set of experiments, a detergent, NP-40, was used, instead of SDS, and no sonication was performed after nuclei purification. Winded histone H3 proteins were not pulled down in the anti-histone H3 immune precipitates. We have provided description of the protocol in "Materials and Methods" section.

Histone H3 is one of the five main histone proteins involved in the structure of chromatin, which showed high sensitivity to epigenetic regulation. Most of histone H3 are winded with genome as nucleosome to maintain chromatin stability, and lesser amounts of H3 are detected as free form or hyper acetylated form (Curr Opin Cell Biol. 2018 Jun;52:126-135.). Acetylation of histone H3 has been shown to unwind with genomic DNA and release to nuclei (Nat Rev Mol Cell Biol. 2017 Feb;18(2):90-101.). Studies have shown that drug treatment (e.g. cocaine) or powerful TF activation induces hyper acetylation of histone H3 and activate 1,696 genes, facilitating hyper acetylation of histone H3 to be decompacted form for gene expression in nuclei (Neuron. 62 (3): 335–48.).

4. *I have a conceptual problem with the orthotopic transplantation experiment in Fig. 4a: The problem with this approach is that tumor cells are not labeled independent of ISX labeling, e.g. by CMV/CAGGS-Luciferase or RFP. Hence, different expression levels of RFP and ISX-RFP might result in different BLI intensities and may be inadequate to monitor tumor cell numbers. Analysis of cryosections of lungs with RFP fluorescence detection would help to clarify this. If not available, at least tumor cells should be analyzed in vitro for RFP fluorescence intensities and heterogeneity by microscopy. Moreover, histological examination/quantification of tumor mass should be done to confirm the results from BLI. Do the tumors also display differences in EMT states? The results in Fig. 4a show that ISX K69 is responsible for efficient tumor growth. However, it cannot be concluded yet whether EMT or metastasis are affected as well.*

Response: We thank for the reviewer's comments. In Fig. 4a, the NSLC cells were RFP-labeled stable NSLC cells infected with lentivirus with different GFP-tagged ISX mutants, which were directly injected into the lung to generate primary tumor mass. We used RFP laser to detect NSLC cell numbers, but not ISX. No signals will be detected if the tumor mass is less than 0.05 centimeter (about 5×10^3 cells). Signals in other organs are indicative of tumor invasion signals through EMT and metastasis. In Fig. 4a, we have made correction on the label in the revised manuscript.

5. *Fig. 4 g+h, mRNA analysis: The results are confusing as they are inconsistent with findings in Fig. 1a. In Fig. 1 a specific effect on Twist and Snail mRNA by ISX ectopic expression alone was observed. Is the observed increase in Fig. 4 due to the combination of ISX/BRD4 on top of the effects of ISX alone? It would be helpful to include mock-transfected samples as well to clarify this. The Figs S2a and 4g+h should be consistent with Fig. 1, showing both Twist and Snail mRNA and protein for the ISX and BRD4-specific experiments.*

Response: We thank for the reviewer's comments. In Figs. 4g and h, in order to verify the effects of BRD4 on the expression of ISX-induced EMT markers, TWIST1 and Snail 1, NSLC cells with ISX expression were transfected with mock (vector only; V-ISX), wild-type and BRD4 mutant (BRD4mutant-ISX), and the value of V-ISX was set as 1. As the Reviewer suggested, we have added another mock control group transfected with reporter constructs expressing GFP and mCherry only in the revised manuscript.

6. *P. 8, Fig. 2d: ISX Co-IP experiment: do the authors claim that in normal lung tissue there is LESS interaction/recruitment found or is there just simply less expression of CBP/CREB, PCAF present to result in weaker Co-IPs of the interaction partners? A specific reduction in interaction can only be claimed for BRD4, since it shows similar expression levels in normal vs. NSCLC, but less interaction to ISX. However, this might be related to the fact that other complex components are less abundant to form a stable ternary/quarternary complex!*

Response: We thank for the reviewer's comments. The Reviewer raised an important issue. The reduction in protein interaction can be as the result of the loss in acetylation or as the result of the lack of other binding proteins recruited to the complex. We cannot discern these possibilities at this time, but the results of our current data would provide a basis for addressing this important question.

7. *Co-IP of the protein complex in Fig. 2c: all analyzed proteins bind with more or less the same extend to each other. Can it be excluded that they are indirectly linked via DNA fragments? The authors should provide another transcription factor/DNA associated protein as negative control which does not bind to most of these IP'd candidates.*

Response: We thank for the reviewer's comments. We have added p53 as a negative control in the new version (Figs. 2 b and S1c).

8. *IF in Fig. 2e+g: BRD4 supposed to be constitutively nuclear, also in A549 cells. The provided IF is not convincing and the discrepancy needs to be discussed. The sites of colocalization/dots are too small and a higher magnification should be included. To verify interaction of the proteins in situ, I suggest to do a Proximity Ligation Assay.*

Response: We appreciate the reviewer's comments. We added the proximity ligation assay (PLA) in the new version as suggested by the reviewer (Figs. 2d and g).

9. *Fig. 4f, ISX-BRD4 co-IP: What is shown in Fig 4f in the blot with the label 'mCherry (Kac)'? Was a lysine acetylation-specific antibody used here? It would be helpful to also show the full blot of the IP including the Mw range of mCherry alone to demonstrate no acetylation of mCherry. This also applies to Fig. 4e (GFP).*

Response: We thank for the reviewer's comments. We have revised the Figs. 4e and f, accordingly and added the MW label in the revised manuscript.

10. *The figure subdivision and distribution needs to be improved. Right now it is a bit random, with one experiment split into two main figures, e.g. Fig. 3i and 4a, Fig. 5a-c belong to Fig. 4.*

Response: We thank for the reviewer's comments. As the Reviewer commented, we revised the Fig. 3i and 4a and Fig. 5a-c to move to one Figure as Figure 4, in new version (see Fig. 3i and j and k; Fig. 4g, h, and i).

11. *Materials and methods miss important experimental procedures, e.g. how the Dox-inducible ISX expressing cell lines have been generated, which antibodies/primer sequences have been used, how ChIP experiments have been performed, etc.*

Response: We thank for the reviewer's comments. We have extensively revised the section of Materials and Methods and figure legends to provide more detailed protocol in various experimental settings.

Minor points:

1. *P. 6: By describing the ChIP experiments in Fig. 1g,h the authors state that the 'aforementioned region showed no or lesser promoter activity than that with GFP expression control'. How can the DNA binding affinity report on the activity of the promoter and what is the GFP control in this experiment?*

Response: We thank for the reviewer's comments. As the Reviewer commented, we have revised the sentence and changed to "ISX-GFP showed high binding activity to the above promoter

elements of both Snail1 and TWIST1 with ISX cis-binding element (IBE; the sequence “CGCCGC” is a potential ISX binding cis-element; [17, 19]) in lung cancer cells (Figures 1e and S1a).” in new version.

2. *P.6. The 'IBE' abbreviation is used before it was defined one sentence later.*

Response: We thank for the reviewer’s comments. As the Reviewer commented, we have added the sentence “IBE [-CGCCGC-] is a potential ISX binding cis-element and found in several promoters of ISX downstream genes” to define “IBE” and added relevant citations in the revised version.

3. *Frequently, the authors use the term 'significant' though no quantification or statistical analyses have been conducted, e.g. in Western blots of S1b on p.9 or Fig. 4e, p. 12.*

Response: We thank for the reviewer’s comments. As the Reviewer commented, we have revised these sentences to avoid confusion.

4. *Molecular weight markers are lacking in all Western blots. This is of particular importance for the IPs that show GFP-fusions vs. GFP-tag only in comparison.*

Response: We thank for the reviewer’s comments. As the Reviewer commented, we have added MW label on immune blotting in the new version.

5. *Fig. 3g axis label: How is 'D/T' in terms of migration activity defined? Why is migration not displayed as in Fig. 2?*

Response: We thank for the reviewer’s comments. In order to improve the clarity, we have revised migration activity in Fig. 3g into the same format in Fig. 2 in the new version.

Referee #3:

The study submitted by Wang et al. identifies PCAF/ISX/BRD4 complex as an activator of Epithelial to Mesenchymal Transition. The authors highlight new targets of PCAF to induce EMT : PCAF acetylates ISX and BDR4 to induce a nuclear translocation of the complex that binds to the promoters of EMT regulators to activate transcription. Overall, data presented is convincing but figures legends are confusing, incomplete and need to be modified. The typography of the figures needs also to be streamlined. All details of the protein interactions are well described. The acetylation of ISX and BRD4 by PCAF appears crucial for EMT induction in NSCLC cellular models and for metastasis formation in mice. The PCAF-ISX-BDR4 axis seems to be also correlated to patient with NSCLC prognosis.

EMT study is a very competitive field of first interest in cancer biology. The study is interesting and follows along a similar line of previous papers. As mentioned in the study, the role of PCAF and BDR4 have already been shown in EMT. In this way, the authors might have referenced a recent study published in CDD showing the involvement of BDR4 in SNAIL regulation at a transcriptional and protein stability levels in breast cancer (Cell Death Differ. 2019, doi: 10.1038/s41418-019-0353-2). The major novelty presented in this study is the implication of ISX in the PCAF/BDR4 translocation in NSCLC cancer to induce EMT.

Based on that some points that need to be addressed or discussed before considering the publication.

Response: We thank for the reviewer’s comments. As the Reviewer commented, we have added the citation “Cell Death Differ. 2019 May 21. doi: 10.1038/s41418-019-0353-2.” as new reference in the introduction section of new version.

Major comments :

1. *How many replicates are performed in western blotting experiments presented Figure 1b : (same remarks for all WB and IP experiments)?*

Response: We thank for the reviewer's comments. We added the number of replications (N=3) for immune blotting and IP analyses in the section of Materials and Methods and Figure legends in all experiments including Figure 1b as the reviewer suggested.

2. I do not understand why the panel Figure 2a is not presented in the Figure 1. Invasion assays are complementary to migration assays and are not in adequacy with the title of the Figure 2 (« Effect of PCAF inhibitor on ISX-BRD4 complex formation, wound healing, and invasive activity »).

Response: We thank for the reviewer's comments. We have revised the Fig. 2a and moved to Fig. 1 in the revised manuscript.

3. The immunofluorescence presented in Figure 2e and f are not convincing. Staining of ISX, BRD4 and PCAF are diffuse in the cells forcing the co-localizations. Proximity Ligation Assay experiments would be more obvious.

Response: We appreciate the reviewer's comments. We added the new data of proximity ligation assay (PLA) to replace the old data in the new version (please see Fig. 2d and g).

4. You present the expression profile of ISX, PCAF and BRD4 in supplemental data, the expression of these proteins seems to be correlated in NSCLC by contrast with WI38. What conclusion do you draw? Is there a common mode of regulation of these proteins/ARNm?

Response: We thank for the reviewer's comments. Analysis of the protein expression levels of ISX, PCAF, and BRD4 in NSLC cells showed the existence of these three components as a functional complex, which correlated with malignance of NSLC cells. WI38 is a CMV-transformed benign lung epithelial cell line and used as a negative control. The NSLC cells with higher levels of protein expression of ISX, PCAF, and BRD4 were more malignant in oncologic activity, and chronic inflammation seems to regulate their protein and mRNA expression.

5. You used different acetyl transferase inhibitors : Tip60 inhibitor, PCAF, P300 inhibitors, GCN5 and you obtained the same effects in EMT inhibition when ISX is overexpressed. How you claim (in the title of this part) that PCAF modulates EMT induction? According to you this phenomenon is specific of PCAF or it exists a redundant function with other acetyltransferases? The specificity of these types of compounds can be discussed, can you use specific siRNAs?

Response: We appreciate the reviewer's comments. The results of Figure 2f indicated that various inhibitor of acetyltransferases such as p300/PCAF, P300/CBP, GCN5 suppressed the expressions of EMT marker proteins (Fig. 2f in old version) and only Garcinol affected the migration and invasion (Fig. 2i and j in Old version). To examine the specificity of PCAF, we used shRNA against PCAF and examined the expression. As suggested by the reviewer, we used shRNA against PCAF and found that the expression of EMT marker genes were inhibited in the presence of ISX-GFP (see Fig. 2g and Supplementary Figure S1i in new version).

6. Why do you perform an IP H3 (Figure 3f) to see the level of histones acetylation ? A western blotting after acid extraction of histones or an ELISA directed against these marks would have enough?

Response: We thank for the reviewer's comments. In Fig. 3, the immune precipitates of anti-histone H3 were used to determine whether or which acetylated lysine was critical for downstream gene expression induced by PCAF-ISX-BRD4 complex. We hypothesized the activation of ISX downstream genes was induced by the H3 acetylation which led nucleosome to be decompacted and facilitated acetylated H3 to be detected in the nuclei. The immune precipitates of anti-histone H3 enriched the acetylated H3 from free form. Immune blotting signals from acid extraction sometime will be criticized whether the signals is targets acetylated or not when none sense signals were also detected in blotting. ELISA detection of acetylated histone H3 was more costly than IP.

7. I do not understand why the panel a and b of the Figure 4 is not presented in the Figure 3. These two panels are showing effects of the absence of acetylation of ISX in tumor grow and are complementary to the panel i of the Figure 3.

Response: We thank for the reviewer's comments. We have moved Figs 4a and b to Figure 3 to generate new version of Figure 3j and k, based on the reviewer's suggestion.

8. *The results obtained Figure 4a show the importance of ISX acetylation in tumor metastasis formation, can you validate this acetylation in your cohort of human tissues ?*

Response: We thank for the reviewer's comments. As the Reviewer commented, we have detected the acetylated ISX level in eight paired NSCLC tumor samples and added the results in Fig.7f in the revised manuscript.

9. *Figure 7 b you cannot speak about « interaction » between ISX, BRD4 and PCAF based on the IF study but just co-localization. In the same way, you never demonstrate in my opinion, the direct interaction between the 3 proteins, or at least a complex formation including these 3 partners. To demonstrate the complex formation, you can perform a native gel for example with 1, 2 or the 3 proteins in the same sample or pull down experiments with these three purified proteins.*

Response: Thank you very much for your kind criticism and we totally agreed with your points. In Figure 7b and c; we used the word of "co-localization" not interaction between ISX, BRD4 and PCAF because they are analyzed by only the IF studies as you criticized. The direct interaction to form the complex with these three proteins was performed several times. But on the native gel we could identify the aggregates complex even in the single protein. Thus, it might be impossible to show the presence of the indicated complex.

Moreover, the purification of these three proteins biochemically is indeed difficult for us because of the spontaneous degradation, especially PCAF. Thus, unfortunately we cannot demonstrate the evidence of the complex formation of these three proteins using the purified proteins. Eventually, we cannot demonstrate these three proteins indeed interacted to each other and for a endogenous complex in cells if we detect these proteins by WB after co-culturing three proteins and pulled then down specifically *in vitro*.

10. *You clustered your NSCLC patients in Low and High expression of ISX. Can you determine the EMT status of your tissues using Vimentin staining as performed in these two studies (Vimentin expression predicts the occurrence of metastases in non-small cell lung carcinomas., doi: 10.1016/j.lungcan.2013.03.011 and EMT is associated with an epigenetic signature of ECM remodeling genes, doi: 10.1038/s41419-019-1397-4). This experiment could highlight a real correlation between your protein complex and EMT in patients with NSCLC.*

Response: Thank you so much for the reviewer's kind suggestion. As review's suggestion, we evaluated the acetylated ISX, Vimentin and EMT status in eight paired NSCLC cell lines (Fig.7f). The expressions of acetylated ISX and vimentin are highly correlated, indicating the EMT status of our samples.

11. *Your final schematic diagram (Fig S3) needs to be modified: The representation of histones acetylation with the elimination of acetylated histone core from the DNA is naive (acetylation of histones leads just to chromatin decompaction). You detailed finely the acetylation sites of your proteins of interest by PCAF in the paper, can you add some details in your scheme (acetylated position...). The global design of your scheme needs to be refined.*

Response: We appreciate the reviewer's suggestions. As the Reviewer commented, we have revised the schematic diagram and added acetylated position of both ISX, BRD4 and histone H3.

Minor comments :

1. *The abbreviation IBE needs to be defined before when you are presenting the IBE-containing region.*

Response: We thank for the reviewer's comments. As the Reviewer commented, we have added the sentence "IBE [-CGCCGC-] is a potential ISX binding cis-element and found in several promoters of ISX downstream genes." to define "IBE" and added relevant citations in the revised version.

2. *ChIP experiments (legend 1) are not described in Mat & Met as mentioned.*

Response: We thank for the reviewer's comments. We have added the relevant information for ChIP and ChIP-ChIP in the Materials and Methods.

3. *Co-IP-2D analysis is not presented in the Mat & Met.*

Response: We thank for the reviewer's comments. We have added the relevant information for Co-IP-2D analysis in the Materials and Methods.

4. *Figure 1g-h : These panels can be placed in supplemental data (not essential, complementary to the panels c and d).*

Response: We thank for the reviewer's comments. As review's suggestion, we moved Figure 1g-h to Fig. S1b in revising new text version.

5. *Figure 1c-d : Homogenize the presentation. Place -200 to +35 at the bottom, before d75-to -89 (as -90 to + 40bp (d87-71bp) panel d).*

Response: We thank for the reviewer's comments. As the Reviewer suggested, we have revised Figure 1c-d in the revised version.

6. *Figure 1i : Explain ISXi in the legend.*

Response: We thank for the reviewer's comments. As the Reviewer commented, we have revised figure legend of Figure 1i in the revised version.

7. *Figure 2 legend: Points f and e : There is a mistake in the figure legend, e and f need to be changed.*

Response: We thank for the reviewer's comments. As the Reviewer commented, we have revised figure legend of Figure 2 in the revised manuscript.

8. *Figure 2clegend: I do not understand why abbreviations of ISX, BRD4, PCAF, P30/CBP, EMT appear at the end of the legend.*

Response: We thank for the reviewer's comments. As the Reviewer commented, we have revised figure legend of Figure 2 in the revised manuscript.

9. *Figure 2f comment « the expression of ISX and most BRD4 showed a significant cytosol-restricted pattern » replace expression by localization.*

Response: We thank for the reviewer's comments. As following to review's suggestion, we revised the description of Figure 2f in revising version.

10. *Supplemental data 1 : There is an empty box before the first amino acid, what is its role?*

Response: We thank for the reviewer's comments. As the Reviewer commented, we have removed the empty box in the revised manuscript.

11. *You need to mention TH1834 as a Tip60 inhibitor in the text (as mentioned for the other compounds)*

Response: We thank for the reviewer's comments. We added the description of TH1834 in revising manuscript.

12. *Figure 2i-j : Streamline the presentation and add GFP after ISX if its presents: GFP/ vehicle, ISX/vehicle and ISX/Garcicol panel i and GFP/V, ISX-GFP/V and ISX/Garnicol. In the same way, for all protein overexpressed mentioned the name of the tag after the name of the protein of interest (GFP, HA, Cherry...) for a better understanding (for example Figure 2i-j, 3g-h...).*

Response: We thank for the reviewer's comments. As the Reviewer commented, we have revised the description in the revised version.

13. Figure 3 a : why do you mentioned the schematic representation of EZH2 (H and M) ? It is not indicated in the legend and on the core of the paper (same comment for Figure 4c).

Response: We thank for the reviewer's comments. We have explained as below. EZH2 (enhancer of zeste homolog 2) is the functional enzymatic component of the Polycomb Repressive Complex 2 (PRC2) and is a histone-lysine N-methyltransferase enzyme (EC 2.1.1.43). Moreover, EZH2 is the first TF proposed to be acetylated by PCAF (Nucleic Acids Res. 2015 Apr 20;43(7):3591-604.). In the previous study we knew that ISX was acetylated by PCAF, but how to identify the acetylation site was the next problem. The study of conserved amino acids sequence of PCAF targets was firstly screened and reported on histone H3, but so far no information is present in ISX. Using our preliminary structural prediction, we predicted the potential acetylated sites of PCAF on EZH2 and ISX, and then by studies of LC mass-mass and point mutation indicated that the real acetylation site on ISX by PCAF. This is a reason that we described the EZH2 in Fig. 3a which might be the target of PCAF as similar to the case of PCAF acetylation site on ISX.

14. Figure 3f: The sentence: « ISX interaction with histone H3 », is not correct. You are only showing the acetylation level of histones after overexpression of ISX WT and mutants.

Response: We thank for the reviewer's comments. We have revised the figure legend to "Acetylation status of histone H3 precipitated by anti-H3 antibody in A549 cells with ectopic ISX or mutants" in the revised version.

15. Figure 4a : B and A at the right of the panel need to be eliminated. Streamline your groups (RFP, ISX-RFP, ISX Ac-RFP or ISX Ac3-RFP).

Response: We thank for the reviewer's comments. As the Reviewer commented, we have revised the description in revised version.

16. Figure 4f: Why AC3 construct is not presented in the figure?

Response: We thank for the reviewer's comments. According to results in previously studies, AC3 mutant showed the same effect as AC1 and 2, so we did not include it in this figure.

2nd Editorial Decision

6 November 2019

Thank you for the submission of your revised manuscript to our editorial offices. We have now received the reports from the three referees that were asked to re-evaluate your study, you will find below. As you will see, the referees now support the publication of your study in EMBO reports. Referee #3 has some remaining points and suggestions to improve the manuscript I ask you to address in a final revised version of the manuscript.

Further, I have these editorial requests:

- Per journal policy, we do not allow 'data not shown' (see page 11 of your manuscript). All data referred to in the paper should be displayed in the main or Expanded View figures, or the Appendix. Thus, please add these data (or change the text accordingly, if these data are not important). See: <http://www.embopress.org/page/journal/14693178/authorguide#unpublisheddata>

- Please call out the figure panels sequentially. Presently, Fig 2G is called out before 2F. Please change this, or the order of the figure panels.

- There is presently no callout for Fig 3H. Please check.

- Please provide clearly visible scale bars for all microscopic images (those in Fig. 1H are not well visible; Fig. 7A still lacks scale bars). Please provide these s uniform single lines without any

writing on them. Please specify the size in the respective figure legend.

- Please check that regarding data quantification and statistics the number "n" for how many independent experiments (biological replicates) were performed, the bars and error bars (e.g. SEM, SD) and the test used to calculate p-values is specified in the respective figure legends. Please provide statistical testing where applicable. See:

<http://www.embopress.org/page/journal/14693178/authorguide#statisticalanalysis>

- Please supply an ORCID ID for the co-corresponding author Shen-Nien Wang. Please find instructions on how to link the ORCID ID in our manuscript tracking system in our author guidelines: <http://www.embopress.org/page/journal/14693178/authorguide#authorshipguidelines>

- Please find attached a word file of the manuscript text (provided by our publisher) with changes we ask you to include in your final manuscript text, and some queries, we ask you to address. Please provide your final manuscript file with track changes, in order that we can see the modifications done.

In addition I would need from you:

- a short, two-sentence summary of the manuscript
- two to three bullet points highlighting the key findings of your study

REFEREE REPORTS

Referee #1:

I have no further questions for this manuscript.

Referee #2:

In the revised version of the manuscript by Hsu and colleagues, the authors extended the analysis by additional experiments and worked on the text. All my points were addressed adequately and in general the new findings significantly improved the quality of the manuscript. I congratulate the authors to their nice work and recommend the manuscript for publication in EMBO Rep.

Referee #3:

The new version of the study answered the majority of my questions. The new experiments performed are very convincing such as PLISA. I would have preferred IHC rather than WB to show the VIMENTIN correlation with ISX and ISX acetylation but the results remain convincing. Figure and legend comments have been improved, however the final schematic diagram (Fig S3) remains approximate and is not on the level of the rest of the study.

Based on the quality of the study and the interest of EMT in cancer biology in my opinion the paper needs to be considered for publication in EMBO reports.

I have just some small comments:

- The M&M section lacks the protocol for the cytoplasm/nuclear extractions.
- In my opinion EZH2 should not be considered as a transcription factor, but as histone methyltransferase or as a "histone writer".
- As seen in Fig 6b, $\Delta B1$ induces also a decrease in DNA binding activity at the Snail promoter as $\Delta B2$. This needs to be mentioned.

2nd Revision - authors' response

8 November 2019

Referee #3:

The new version of the study answered the majority of my questions. The new experiments performed are very convincing such as PLISA. I would have preferred IHC rather than WB to show the VIMENTIN correlation with ISX and ISX acetylation but the results remain convincing. Figure and legend comments have been improved, however the final schematic diagram (Fig S3) remains approximate and is not on the level of the rest of the study.

Based on the quality of the study and the interest of EMT in cancer biology in my opinion the paper needs to be considered for publication in EMBO reports.

I have just some small comments:

- The M&M section lacks the protocol for the cytoplasm/nuclear extractions.

Response: We thank for the reviewer's comments. We have added the description of for cytoplasm/nuclear extractions method in the "Materials & Methods" section.

- In my opinion EZH2 should not be considered as a transcription factor, but as histone methyltransferase or as a "histone writer".

Response: We thank for the reviewer's comments. We have removed "EZH2" out of the sentence.

- As seen in Fig 6b, $\Delta B1$ induces also a decrease in DNA binding activity at the Snail promoter as $\Delta B2$. This needs to be mentioned

Response: We thank for the reviewer's comments. We have added it in the new manuscript.

Acceptance

22 November 2019

I am very pleased to accept your manuscript for publication in the next available issue of EMBO reports. Thank you for your contribution to our journal.

Corresponding Author Name: Shih-Hsien Hsu

Journal Submitted to: EMBO REPORTS

Manuscript Number: EMBOR-2019-48795V1